# SHAPE-TAILORED DEEP NEURAL NETWORKS

## ABSTRACT

We present Shape-Tailored Deep Neural Networks (ST-DNN). ST-DNN extend convolutional networks (CNN), which aggregate data from fixed shape (square) neighborhoods, to compute descriptors defined on arbitrarily shaped regions. This is natural for segmentation, where descriptors should describe regions (e.g., of objects) that have diverse shape. We formulate these descriptors through the Poisson partial differential equation (PDE), which can be used to generalize convolution to arbitrary regions. We stack multiple PDE layers to generalize a deep CNN to arbitrary regions, and apply it to segmentation. We show that ST-DNN are covariant to translations and rotations and robust to domain deformations, natural for segmentation, which existing CNN based methods lack. ST-DNN are 3-4 orders of magnitude smaller then CNNs used for segmentation. We show that they exceed segmentation performance compared to state-of-the-art CNN-based descriptors using 2-3 orders smaller training sets on the texture segmentation problem.

## 1 INTRODUCTION

Convolutional neural networks (CNNs) have been used extensively for segmentation problems in computer vision He et al. (2017); He et al. (2016); Chen et al. (2017); Xie & Tu (2015). CNNs provide a framework for learning descriptors that are able to discriminate different textured or semantic regions within images. Much progress has been made in segmentation with CNNs but results are still far from human performance. Also, significant engineering must be performed to adapt CNNs to segmentation problems. A basic component in the architecture for segmentation problems involves labeling or grouping dense descriptors returned by a backbone CNN. A difficulty in grouping these descriptors arises, especially near the boundaries of segmentation regions, as CNN descriptors aggregate data from fixed shape (square neighborhoods) at each pixel and may thus aggregate data from different regions. This makes grouping these descriptors into a unique region difficult, which often results in errors in the grouping.

In segmentation problems (e.g., semantic segmentation), current methods attempt to mitigate these errors by adding post-processing layers that aim to group simultaneously the (coarse-scale) descriptors from the CNN backbone and the fine-level pixel data. However, the errors introduced might not always be fixed. A more natural approach to avoid this problem is to consider the coarse and fine structure together, avoiding aggregation across boundaries, to prevent errors at the outset.

To avoid such errors, one could design descriptors that aggregate data only within boundaries. To this end, Khan et al. (2015) introduced "shape-tailored" descriptors that aggregate data within a region of interest, and used these descriptors for segmentation. However, these descriptors are hand-crafted and do not perform on-par with learned approaches. Khan & Sundaramoorthi (2018) introduced learned shape-tailored descriptors by learning a neural network operating on the input channel dimension of input hand-crafted shape-tailored descriptors for segmentation. However, these networks, though deep in the channel dimension, did not filter data spatially within layers. Since an advantage of CNNs comes from exploiting spatial filtering at each depth of the network, in this work, we design shape-tailored networks that are deep and perform shape-tailored filtering in space at *each* layer using solutions of the Poisson PDE. This results in shape-tailored networks that provide more discriminative descriptors than a single shape-tailored kernel. This extension requires development of techniques to back-propagate through PDEs, which we derive in this work.

Our contributions are specifically:

1. We construct and show how to train ST-DNN, deep networks that perform shape-tailored spatial filtering via the Poisson PDE at each depth so as to generalize a CNN to arbitrarily shaped regions.
2. We show analytically and empirically that ST-DNNs are covariant to translations and rotations as they inherit this property from the Poisson PDE. In segmentation, covariance (a.k.a., equivariance)

to translation and rotation is a desired property: if a segment in an image is found, then the corresponding segment should be found in the translated / rotated image (or object). This property is not generally present with existing CNN-based segmentation methods even when trained with augmented translated and rotated images Azulay & Weiss (2019), and requires special consideration.

3. We show analytically and empirically that ST-DNNs are robust to domain deformations. These result from viewpoint change or object articulation, and so they should not affect the descriptor.

4. To demonstrate ST-DNN and the properties above, we validate them on the task of segmentation, an important problem in low-level vision Malik & Perona (1990); Arbelaez et al. (2011b).

Because of properties of the PDE, ST-DNN also have desirable generalization properties. This is because: a) The robustness and covariance properties are built into our descriptors and do not need to be learned from data, b) The PDE solutions, generalizations of Gabor-like filters Olshausen & Field (1996); Zador (2019), have natural image structure inherent in their solutions and so this does not need to be learned from data, and c) Our networks have fewer parameters compared to existing networks in segmentation. This is because the PDE solutions form a basis and only linear combinations of a few basis elements are needed to learn discriminative descriptors for segmentation. In contrast, CNNs spend a lot of parameters to learn this structure.

## 1.1 RELATED WORK

Traditional approaches to segmentation rely on hand-crafted features, e.g., through a filter bank Haralick & Shapiro (1985). These features are ambiguous near the boundaries of objects. In Khan et al. (2015) hand-crafted descriptors that aggregate data within object boundaries are constructed to avoid this, but lack sufficient capacity to capture the diversity of textures or be invariant to nuisances. Deep-learning based approaches have showed state-of-the-art results in edge-based methods Xie & Tu (2017); He et al. (2019); Deng et al. (2018). Watershed is applied on edge-maps to obtain the segmentation. The main drawback of these methods is it is often difficult to form segmentations due to extraneous or faint edges, particularly when "textons" in textures are large.

CNNs have been applied to compute descriptors for semantic segmentation, where pixels in an image are classified into certain semantic object classes Li et al. (2019); Huang et al. (2019); Du et al. (2019); Pang et al. (2019); Zhu et al. (2019); Liu et al. (2019). Usually these classes are limited to a few object classes and do not tackle general textures, where the number of classes may be far greater, and thus such approaches are not directly applicable to texture segmentation. But semantic segmentation approaches may eventually benefit from our methodology as descriptors aggregating data only within objects or regions are also relevant to these problems. A learned shape-tailored descriptor Khan & Sundaramoorthi (2018) is constructed with a Siamese network on hand-crafted shape-tailored descriptors. However, Khan & Sundaramoorthi (2018) only does shape-tailored filtering in pre-processing as layering these requires new methods to train. We further examine covariance and robustness, not examined in Khan & Sundaramoorthi (2018).

Covariance to rotation in CNNs has been examined in recent works, e.g., Weiler et al. (2018); Yin et al. (2019); Anderson et al. (2019). They, however, are not shape-tailored so do not aggregate data only within shaped regions. Lack of robustness to deformation (and translation) in CNNs is examined in Azulay & Weiss (2019) and theoretically in Bietti & Mairal (2017). Sifre & Mallat (2013) constructs deformation robust descriptors inspired by CNNs, but are hand-crafted.

## 2 CONSTRUCTION OF SHAPE-TAILORED DNN AND PROPERTIES

In this section, we design a deep neural network that outputs descriptors at each pixel within an arbitrary shaped region of interest and aggregates data only from within the region. We want the descriptors to be discriminative of different texture, yet robust to nuisances within the region (e.g., local photometric and geometric variability) to be useful for segmentation. Our construction uses a Poisson PDE, which naturally smooths data only within a region of interest. Smoothing naturally yields robustness to geometric nuisances (domain deformations). By taking linear combinations of derivatives of the output of the PDE, we can approximate the effect of general convolutional kernels but avoid mixing data across the boundary of region of interest. ST-DNN is also covariant to

translations and rotations, inheriting it from the Poisson equation, which leads to the segmentation algorithm being covariant to such transformations.

## 2.1 SHAPE-TAILORED DNN DESCRIPTORS THROUGH POISSON PDE

**Shape-tailored Smoothing via Poisson PDE**: To construct a shape-tailored deep network, we first smooth the input to a layer using the Poisson PDE so as to aggregate data only within the region of interest, similar to what is done in Khan et al. (2015) for just the first layer. Let $R \subset \Omega \subset \mathbb{R}^2$ be the region of interest, where $\Omega$ is the domain of the input image $\mathbf{I} : \Omega \to \mathbb{R}^k$ and $k$ is the number of input channels to the layer. Let $\mathbf{u} : R \to \mathbb{R}^M$ ($M$ is the number of output channels) be the result of the smoothing; the components $u$ of $\mathbf{u}$ solve the PDE within $R$:

$$\begin{cases} u(x) - \alpha \Delta u(x) = I(x) & x \in R \\ \nabla u(x) \cdot N = 0 & x \in \partial R \end{cases}, \qquad (1)$$

where $I$ is a channel of $\mathbf{I}$, $\partial R$ is the boundary of $R$, $N$ is normal to $\partial R$, $\alpha$ is the scale of smoothing and $\Delta / \nabla$ are the Laplacian and the gradient respectively. It can be shown that the smoothing can be written in the form $u(x) = \int_R K(x, y) I(y) dy$ where $K(.,.)$ is the Green's function of the PDE, a smoothing kernel, which further shows that the PDE aggregates data only within $R$.

**Shape-tailored Deep Network**: We can now generalize the operation of convolution tailored to the region of interest by taking linear combinations of partial derivatives of the output of the PDE equation 1. This is motivated by the fact that in $R = \mathbb{R}^2$, linear combinations of derivatives of Gaussians can approximate any kernel arbitrarily well. Gaussian filters are the solution of the heat equation, and the PDE equation 1 relates to the heat equation, i.e., equation 1 is the steady state solution of a heat equation. Thus, linear combinations of derivatives of equation 1 generalize convolution to an arbitrary shape $R$; in experiments, a few first order directional derivatives are sufficient for our segmentation tasks (see Section 5 for details). A layer of the ST-DNN takes such linear combinations and rectifies it as follows:

$$f_i(x) = r \circ L_i \circ T[I](x), \qquad (2)$$

where $I : \mathbb{R} \to \mathbb{R}^k$ is the input to the layer, $T$ is an operator that outputs derivatives of the solution of the Poisson PDE equation 1, $L_i(y) = w_i y + b_i$ is a point-wise linear function (i.e., a $1 \times 1$ convolution applied to combine different channels), $r$ is the rectified linear function, and $i$ indexes the layer of the network. Notice that since $r$ and $L_i$ are pointwise operations, they preserve the property of $T$ that it aggregates data only within the region $R$. We now compose layers to construct a ST-DNN as follows:

$$F[I](x) = s \circ f_m \circ f_{m-1} \circ f_{m-2} \circ .... f_0 \circ I(x), \qquad (3)$$

where $F[I](x)$ is the output of the ST-DNN, $f_0, ..., f_m$ are the $m + 1$ layers of the network, $I$ is the input image, and $s$ represents the soft-max operation (to bound the output values).

ST-DNN does not have a pooling layer because the PDE already aggregates data from a neighborhood by smoothing; further, the lack of reduction in spatial dimension allows for more accurate shape estimation in our subsequent segmentation, and avoids the need for up-sampling layers. We will show that we can retain efficiency in training and inference.

## 2.2 COVARIANCE AND ROBUSTNESS OF ST-DNN

In addition to ST-DNN generalizing CNNs to arbitrary shaped regions, the ST-DNN is also covariant to in-plane translation and rotation, and robustness to domain deformations due to properties of the Poisson PDE. This means covariance also extends to our segmentation method, which is important as any object segmented in an image will also be segmented if the camera undergoes these transformations. Robustness to deformations is important as this means that small geometric variability (e.g., shape variations in textons, small viewpoint change, object deformation) will not affect the descriptors and the segmentation much. We make these properties more precise, and give intuition for proofs, leaving details to Appendix B.

A covariant operator commutes with a set of transformations:

**Definition 1** *An operator $S : \mathcal{I} \to \mathcal{I}$ (from the set of images $\mathcal{I}$ to itself) is **covariant** to a class $\mathcal{W}$ of transformations if $S[I \circ w] = [SI] \circ w$ for every $I \in \mathcal{I}$ and $w \in \mathcal{W}$.*

ST-DNN is covariant to $\mathcal{W}$, the set of in-plane rotations and translations:

**Theorem 1** *The ST-DNN equation 3 is covariant to the set of translations and rotations, i.e., $x \rightarrow \mathcal{R}x + \mathcal{T}$ where $\mathcal{R}$ is a $2 \times 2$ rotation matrix and $\mathcal{T} \in \mathbb{R}^2$.*

This follows from the covariance of the Laplacian and point-wise operations (rectification, $1 \times 1$ convolution), and lack of sub-sampling.

We now make precise the robustness of the ST-DNN to domain deformations:

**Theorem 2** *The ST-DNN equation 3 is insensitive to deformations, i.e.,*

$$|F[I \circ w] - F[I]| \leq C\|w - id\|_{H^1}, \tag{4}$$

*where $w : \Omega \rightarrow \Omega$ is a domain deformation, $id$ is the identity map, $H^1$ is the Sobolev norm (measures both the amount and smoothness of the deformation), and $C$ is a constant independent of $w, I$.*

Intuitively, this follows from the fact that the Poisson PDE locally averages input data, and local averages are robust to translation and hence deformations, which are locally translations.

## 3 TRAINING OF THE NETWORK AND BACK-PROPAGATION

In this section, we describe the training of ST-DNN by introducing a loss function, how the weights can be learned, and the implementation. As the network layers solutions to PDEs, one needs to differentiate through such layers, which we describe.

### 3.1 LOSS FUNCTION FOR TRAINING

Given the ST-DNN of Section 2.1, the loss function to train such descriptors from ground truth segmentation masks (motivated by consistency to the segmentation algorithm in Section 4 that is based on classical energies Chan & Vese (2001); Yezzi Jr et al. (1999) from computer vision) is defined as:

$$L(\mathbf{W}) = \sum_{i=1}^{N} \frac{1}{|R_i|} \int_{R_i} ||\mathbf{F_W}(x) - \mathbf{a_i}||_2^2 dx - \sum_i \sum_{j \neq i} ||\mathbf{a_i} - \mathbf{a_j}||_2^2 \tag{5}$$

where $i, j \in \{1, 2, ..., N\}$ are the indices for the regions in the ground truth segmentation, $\mathbf{F_W}(x)$ is the output of the ST-DNN, $\mathbf{W}$ are the weights of the network (i.e., weights on derivatives of the Poisson PDE solution), $|R_i|$ is the area of region $i$, and $\mathbf{a_i}$ is the average descriptor within the $i^{\text{th}}$ region, i.e., $\mathbf{a_i} = \frac{1}{|R_i|} \int_{R_i} \mathbf{F_W}(x) dx$. The loss function is comprised of two terms. The first component of the loss is minimized when the learned descriptor is constant within regions $R_i$ so that each region consists of parts of the image with uniform descriptor. The second term forces the learned descriptor of different regions to be different to discriminate different textures.

### 3.2 COMPUTING GRADIENTS OF THE LOSS AND TRAINING

Computing gradients of the loss function for training requires consideration as it involves differentiating through PDEs. The most straightforward way to do this involves discretizing the PDE, so the solution is a linear matrix system as we do below. This allows the use of existing deep learning packages to perform back-propagation by storing the matrix in memory. However, this can lead to large memory consumption as the matrix can be large and is only feasible for small images. Fortunately, our PDEs involve a scale parameter $\alpha$ so we can train using down-sampled images, facilitating the use of existing packages for back-prop, and then infer on native resolution images by simply scaling $\alpha$ by the down-sampling factor, which we do in experiments.

The more accurate method, though more difficult to implement, is to avoid storing the matrix and instead compute the PDE solution by an iterative numerical PDE method that does not require storage of matrices. This requires formulating a variant of the back-propagation algorithm, that is similar but involves forward propagation of layer derivatives with respect to the weights through the PDE solutions. This is unfortunately not available in standard deep learning packages. For completeness, we provide the mathematical formulation of this approach in Appendix C,for which

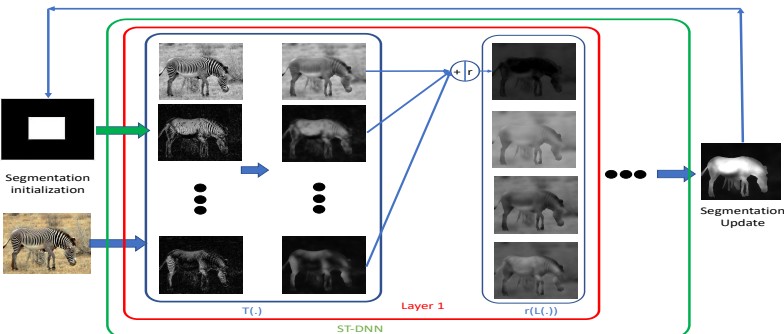

Figure 1: **ST-DNN computation and joint segmentation.** The input to the network is the image and an initial segmentation mask. ST-DNN dense descriptors are computed for each region of the mask using equation 3. The segmentation updates by taking a few steps in the gradient direction of equation 8. The process is iterated with the updated segmentation until the regions converge.

we have performed experiments for few layer cases. This did not give an appreciable performance increase given the complexity of implementation, but could be useful in other applications.

**Implementation**: Using the first method above, we discretize equation 1 as:

$$u(i,j) - \alpha \cdot \sum_{k,l \in \mathcal{N}(i,j) \cap R} [u(k,l) - u(i,j)] = I(i,j), \quad \text{for } (i,j) \in R \tag{6}$$

where $\mathcal{N}(i,j)$ represents the 4-pixel neighborhood of pixel $(i,j)$, which represents the $i^{\text{th}}$ row and $j^{\text{th}}$ column, and intersection means that only neighbors in the region are considered as implied by the Neumann boundary condition in the PDE, avoiding aggregation outside $R$. The discretization approximately preserves the rotation covariance, and any errors vanish with increasing resolution. Note that more accurate discretizations exist, but our experiments demonstrate the sufficiency of this scheme. We can vectorize $u$ and $I$ and write equation 6 as:

$$\mathbf{A_R u = I} \quad \text{and} \quad \mathbf{u = A_R}^{-1}\mathbf{I}. \tag{7}$$

The above is a linear transformation from $\mathbf{I}$ to $\mathbf{u}$. The size of $\mathbf{A}$ is $(mn \times mn)$, where $m$ and $n$ are the number of rows and columns in $I$. With the PDE layers defined through this matrix multiplication, we can use the usual back-propagation method to compute derivatives with respect to weights (see Appendix D for more details). In experiments, we downsampled images to $32 \times 32$ for training.

## 4 APPLICATION TO SEGMENTATION

In this section, we describe the procedure for segmentation using the trained ST-DNN. During inference time, the regions of segmentation are estimated iteratively together with updates of the ST-DNN for each of the regions as they evolve. The evolution of the regions to determine the segmentation is obtained by optimizing the following energy (based on the classical energy Chan & Vese (2001)):

$$E(\mathbf{R}) = \sum_{i=1}^{N} \int_{R_i} \|\mathbf{F_{R_i}}(x;W) - \mathbf{a_i}\|_2^2 dx + \beta \int_{\partial R_i} ds, \tag{8}$$

where $\mathbf{R} = \{R_1, R_2, ..., R_N\}$ are regions in segmentation, $\mathbf{F_{R_i}}$ is the shape-tailored descriptor from the DNN given the learned weights $W$ and within $R_i$, and $\beta > 0$ is the arclength regularization parameter. Note that this energy differs from the loss function used for training in two ways. First, the second term in equation 5 is omitted as it is used in training to avoid the descriptor from learning to be uniform across different textured regions; during inference, the network is already trained to be different across different regions. Second, we add regularization to keep the region boundaries smooth; it is not needed in training since we do not solve for the regions as ground truth is available.

To minimize the (non-convex) energy with respect to the region, we use gradient decent. The gradient with respect to the region $R_i$ is approximately given by $[\|\mathbf{F}_{\mathbf{R_i}}(x; W) - \mathbf{a_i}\|_2^2 - \|\mathbf{F}_{\mathbf{R_j}}(x; W) - \mathbf{a_j}\|_2^2 + \beta \kappa_i] N_i$ where $N_i$ is the unit outward normal to the region, and $\kappa_i$ is its curvature. The curve (boundary of regions) evolution to determine the regions is implemented with a method analogous to level set methods Osher & Sethian (1988) by evolving smooth indicator functions of regions for convenient implementation, details of the algorithm and implementation are shown in Algorithm 2 in Appendix D. The method involves joint updates of the regions and the shape-tailored descriptors within the evolving regions (see Figure 1). We used a box tessellation to initialise the regions, typical of level set methods for segmentation. Our method typically takes a few iterations (approx. 20) to converge in our experiments.

## 5 EXPERIMENTS

**Network Architecture:** We use a 4 layer ST-DNN, which is optimal for the datasets used: fewer layers lead to less accuracy and more layers lead to overfitting (see Appendix E). The layer $f_0$ outputs a 40 dimensional descriptor, with 3 color channels, a gray scale channel and oriented gradients at angles $\{0, \pi/4, \pi/2, 3\pi/4\}$ over 5 shape-tailored smoothing levels (scales) $\alpha = \{5, 10, 15, 20, 25\}$. The four fully-connected layers have 100, 40, 20, 5 hidden units respectively. The smoothing parameter $\alpha$ for all subsequent layers is set to 5. The training on the datasets (below) takes less than 2 hours on Nvidia Quadro RTX 6000 GPU and Intel Xeon 2.60GHz CPU. The inference time (joint segmentation) is 2 seconds on images of size $256 \times 256$.

**Datasets:** We apply ST-DNN to texture segmentation (since covariance and robustness properties are important for texture descriptors Julesz (1981); Sifre & Mallat (2013)). We evaluate on two challenging texture segmentation datasets Khan et al. (2015) - the Real-World Texture Segmentation dataset (RWTSD) - 256 complex real-world images (128 training and testing images) and the Synthetic Texture Dataset consists of 200 test images and 300 training images generated from the Brodatz dataset. We have also tested on multi-region segmentation BSDS500 and Synthetic Texture datasets, details are provided below.

**Methods:** We compare our method against popular deep learning architectures in computer vision - DeepLab-v3 Chen et al. (2017), and FCN-ResNet101 He et al. (2016). In our notation resnet/deeplabv3-x-y, resnet and deeplabv3 represents FCN-Resnet101 and DeepLab-v3 respectively, x denotes the data used in training (subsequently fine-tuned on the texture segmentation dataset). 'x' can be 'm' for MSRA dataset, 'd' for DUTS dataset, 'all' for a combination of all datasets (see Appendix E for more details), and 'TD' for RWTSD (augmented with 8 rotations and 5 scales). 'y' denotes the loss function, 'ce' represents cross-entropy and 'ours' represents the loss introduced in this paper. Segmentation for all methods is done by minimizing equation 8. We also compare our methods against the state-of-the-art methods for texture segmentation, which contain both the classical Arbelaez et al. (2011b); Arbeláez et al. (2014); Isola et al. (2014); Kokkinos (2015); Khan et al. (2015) and deep-learning methods Khan & Sundaramoorthi (2018). ST-DNN is trained only on texture datasets.

**Evaluation Metrics**: We compare on evaluation metrics from Arbelaez et al. (2011a). Ground truth covering (GT-Cov), Random Index (RI) and Variation of Information (VOI) measures region accuracy (higher GT-Cov, RI and lower VOI are more accurate), and F-measure (higher is better) measures boundary accuracy.

**Ablation Studies**: We have performed ablations studies that are summarized in Appendix E Table 4. They show that more PDE filtering layers give higher accuracy (up to a point of overfitting at 4 layers), and that shape-tailored descriptors updated as the region updates outperforms non-shape tailored descriptor (ST-DNN computed on the whole image and not updated as the region evolves).

**Testing Covariance:** To demonstrate the covariance of ST-DNN to translation and rotation, we performed an experiment on Real-world Texture Segmentation dataset. Each image in the test set was randomly rotated with $\theta \in \{\pi/6, 2\pi/6, 3\pi/2, 4\pi/6, 5\pi/6, \pi\}$ and cropped to a rectangle at random positions (to simulate translation) in the rotated image; we ensure the rectangle only contains data from the original image. We segment the original and the transformed image, denoted $S[I]$ and $S[I \circ w]$, respectively, where $w$ is the transformation used to produce the translated/rotated image. We then measure the difference between $S[I] \circ w$ and $S[I \circ w]$ through GT-covering; both should

be equal if the descriptor is covariant. Results are summarized in Figure 2. ST-DNN outperforms resnet101-all-ce by a margin of almost 25%. Note ST-DNN uses no data augmentation, whereas the competing networks are augmented with translated and rotated images from RWTSD. Perfect covariance may not be achieved as translation/rotation also necessarily include occlusion.

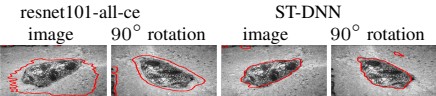

| | ST-DNN | | resnet101-all-ce | |
|---|---|---|---|---|
| | GT Covering | Rand Index | GT Covering | Rand Index |
| | 0.87 | 0.89 | 0.69 | 0.70 |

Figure 2: Comparison of covariance to rotation and translation of ST-DNN, and sota CNN descriptor. Left: A sample result with outputs for original and transformed images. Right: Quantitative result on Real-World Texture Dataset: Higher scores indicate better covariance.

**Testing Deformation Robustness:** To demonstrate robustness to deformation, we apply the trained networks above to randomly deformed versions of the RWTSD test set. We generate random smooth deformations using truncated Fourier series: $v(x) = \sum_{k=-N}^{N} a_k \exp(i2\pi k \cdot x)$ where $x \in [0,1]^2, k = (k_1, k_2)$, $w(x) = x + v(x)$ is the deformation, $a_k$ is randomly generated, and $N = 10$ (appropriate for the resolution). The Soboev norm is $\|v\|^2 = |a_0|^2 + \sum_{k=-N}^{N} |k|^2 |a_k|^2$. For each image in the dataset we generate 8 random deformations of varying norm $\|v\|^2$ from 10 to 80 in steps of 10. We examine the robustness of descriptors to deformations of increasing norm by comparing the segmentation of the original and deformed images similar to the previous experiment. Results and qualitative samples are in Figure 3, which show that ST-DNN is more robust by large margins than competing descriptors, and the robustness over competing methods increases with increasing norm. Note a descriptor could be robust, but not accurate, but this is not the case for ST-DNN (next experiments).

original image (left), deformed images (increasing deformation ⟶)

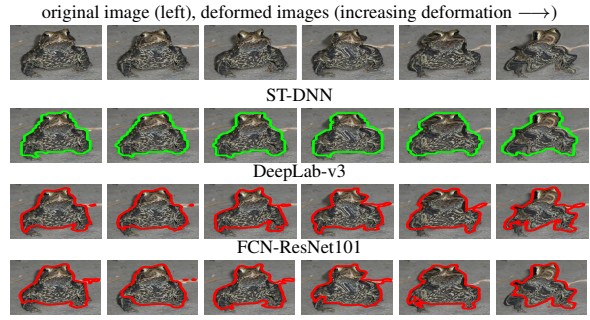

|  | GroundTruth Covering | | |
|---|---|---|---|
| Sobolev Norm | 20 | 40 | 80 |
| DeepLab-v3 | 0.85 | 0.76 | 0.66 |
| FCN-ResNet101 | 0.81 | 0.75 | 0.65 |
| ST-DNN | **0.88** | **0.85** | **0.81** |
|  | Rand Index | | |
| Sobolev Norm | 20 | 40 | 80 |
| DeepLab-v3 | 0.86 | 0.77 | 0.68 |
| FCN-ResNet101 | 0.82 | 0.77 | 0.68 |
| ST-DNN | **0.89** | **0.86** | **0.82** |

Figure 3: *Comparison of robustness to deformations of ST-DNN with sota CNN descriptors. Sample results on segmentation of original and deformed images (left), and quantitative results (right): higher values indicates more robustness.*

**Comparison of ST-DNN to Standard DNNs**: We segment descriptors (ST-DNN and common deep network backbones) by minimizing equation 8. Quantitative results are in Table 1 and qualitative samples are shown in Figure 5. ST-DNN outperforms all other descriptors. ST-DNN has **8900** parameters and is trained with only **128** training images of real world texture dataset. ST-DNN is around 3 orders of magnitude smaller than standard deep networks and takes around 2 orders of magnitude less training data (e.g., FCN-ResNet101-all-ours uses 50,000 images plus augmented data and has 45 million parameters), but still outperforms these networks (see Figure 4).

**Comparison to Texture Segmentation Methods on Real-World & Brodatz Synthetic Texture Datasets:** We compare to state-of-the-art texture segmentation methods: edge-based (gPb, MCG, Kok, CB) Arbelaez et al. (2011b); Arbeláez et al. (2014); Isola et al. (2014); Kokkinos (2015)) and region based (STLD Khan et al. (2015) and Siamese Khan & Sundaramoorthi (2018)). STLD is a shape-tailored (but hand crafted) approach using PDEs, and non-STLD uses the same PDE as STLD but on the whole image so is not shape-tailored. Siamese uses a neural network on the channel dimension returned by STLD, but does not layer PDE solutions as our approach. Quantitative results for both texture datasets are in Table 2 and a few qualitative samples of the results are in Figure 5. Our method out-performs all others by significant margins on all regions metrics and achieves close

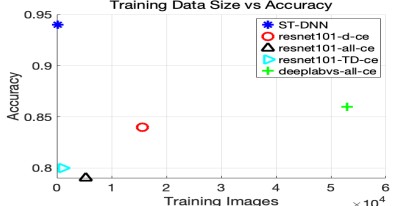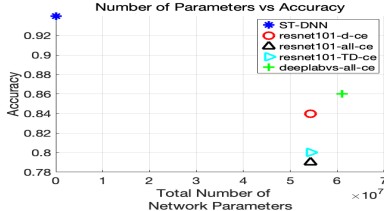

Figure 4: *ST-DNN is smaller in size and uses fewer training images compared with SOTA DNNs.*

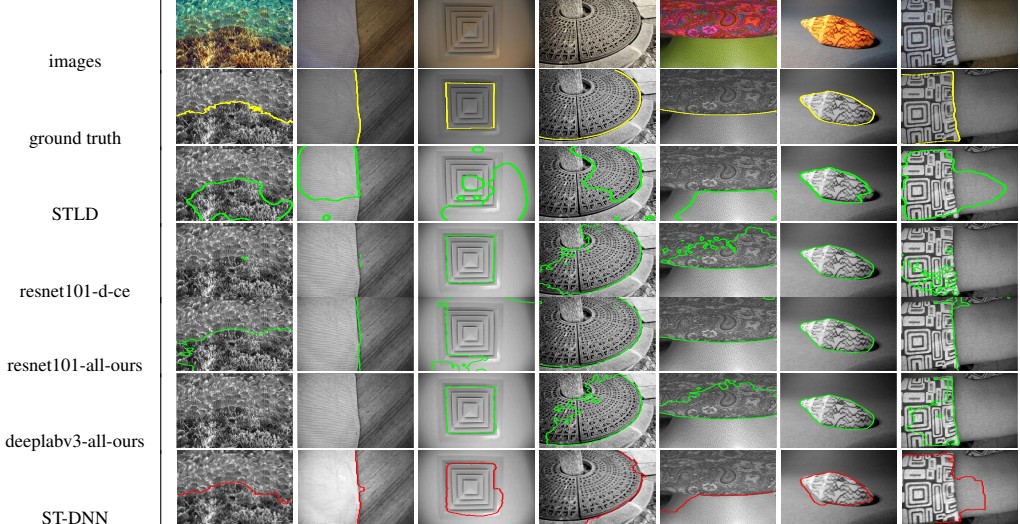

Figure 5: **Sample representative results on Real-World Texture Dataset**. *We compare the ST-DNNs (ours), STLD, and deep learning based methods.*

to the best result on the contour metric on RWTSD, while on the Synthetic Brodatz, our method out performs all methods on all metrics. More visual results and comparison to more methods is in Appendix E.

| | Contour | Region metrics | | |
|---|---|---|---|---|
| | F-meas. | GT-cov. | Rand. Index | Var. Info. |
| ST-DNN (ours) | **0.63** | **0.94** | **0.94** | **0.35** |
| resnet101-d-ce | 0.39 | 0.84 | 0.75 | 0.70 |
| resnet-d-ours | 0.39 | 0.83 | 0.76 | 0.74 |
| resnet101-all-ce | 0.04 | 0.79 | 0.55 | 1.39 |
| resnet101-all-ours | 0.48 | 0.83 | 0.87 | 0.50 |
| resnet101-TD-ours | 0.17 | 0.83 | 0.66 | 0.94 |
| resnet101-TD-ce | 0.11 | 0.80 | 0.63 | 1.35 |
| deeplabv3-all-ce | 0.42 | 0.86 | 0.85 | 0.66 |
| deeplabv3-all-ours | 0.42 | 0.85 | 0.76 | 0.67 |
| HEDXie & Tu (2015) | 0.04 | 0.53 | 0.60 | 1.69 |

Table 1: **Results on Real-World Segmentation Datasets of Deep Networks**. Algorithms are evaluated on contour/ region metrics. Higher F-measure for the contour metric, ground truth covering (GT-cov), and rand index indicate better fit, and lower variation of information (Var. Info) indicates a better fit to ground truth.

**Comparison on Multi-Region Segmentation:** We have also tested our method on two multi-region segmentation datasets, 1) BSDS500 2) Synthetic Texture dataset. For BSDS500 dataset we train our methods and state of the art deep networks on the training + validation set (200+100 images augmented with with 8 rotations and 5 scales) and test on the test-set on BSDS dataset (200 images). Results are summarized in Table 3 and a few quantitative samples are show in figure 9 in Appendix E. BSDS500 is not the best dataset for training on region segmentation as the regions are not marked based on appearance and instead watershed is used on edge maps to generate segments. Hence region with similar looking appearance are often marked as separate regions in images. Despite this drawback ST-DNN performs on par or better than state of the art Deep Neural Networks. During training all images are resized to $256 \times 256$ and normalised to have zero mean and unit variance. In post-processing, the outputs are clustered into 20 regions and then regions with less than 2% pixels

| Real-World Texture Dataset | | | | | | Synthetic Brodatz Texture Dataset | | | | |
|---|---|---|---|---|---|---|---|---|---|---|
| | Contour | Region metrics | | | | | Contour | Region metrics | | |
| | F-meas. | GT-cov. | Rand. Index | Var. Info. | | | F-meas. | GT-cov. | Rand. Index | Var. Info. |
| ST-DNN (ours) | 0.64 | **0.94** | **0.94** | **0.35** | ST-DNN (ours) | **0.49** | **0.92** | **0.92** | **0.44** |
| Siamese | 0.65 | 0.92 | 0.92 | 0.43 | Siamese | 0.45 | 0.90 | 0.89 | 0.46 |
| STLD | 0.58 | 0.86 | 0.88 | 0.63 | STLD | 0.41 | 0.87 | 0.86 | 0.53 |
| mcg | 0.54 | 0.82 | 0.85 | 0.66 | gPb | 0.40 | 0.81 | 0.82 | 0.75 |
| Kok. | 0.64 | 0.56 | 0.57 | 0.92 | CB | 0.30 | 0.77 | 0.79 | 1.09 |
| non-STLD | 0.20 | 0.83 | 0.84 | 0.79 | non-STLD | 0.18 | 0.84 | 0.84 | 0.65 |

Table 2: **Results on Real-World Segmentation Dataset (left) and Synthetic Dataset (right)**. Comparisons are performed against CB Isola et al. (2014), gPb Arbelaez et al. (2011b), Siamese Khan & Sundaramoorthi (2018), Kok.Kokkinos (2015), mcg Arbeláez et al. (2014). See Table 1 caption for details on the measures. Additional results in Appendix E

| BSDS | | | | | Synthetic Multi-Region Texture Dataset | |
|---|---|---|---|---|---|---|
| | Region metrics | | | | | Region Metric |
| | GT-cov. | Rand. Index | Var. Info. | | | Avg. Accuracy |
| Deeplab-V3 | **0.39** | 0.72 | 1.88 | Deeplab-v3 | | 0.41 |
| FCN-Resnet | 0.38 | 0.71 | 1.98 | FCN-resnet101 | | 0.43 |
| ST-DNN(ours) | **0.39** | **0.73** | **1.80** | ST-DNN (ours) | | **0.45** |

Table 3: **Results on BSDS500 (left) and Synthetic multi-region Dataset (right)**. Comparisons are performed against state-of-the-art deep learning based methods. See Table 1 caption for details on the measures

.

of the entire image are smoothed out using conditional random fields, we use pydensecrf [1] library. Following which we run 20 iterations of 2 . For quantitative comparisons we have provided analyses on region metrics on BSDS500 benchmark.

We have also tested on a large scale multi-region synthetic texture segmentation dataset [2]. This dataset is designed as an extension of Khan et al. (2015), we have a maximum of 4 regions per image comprising of different appearance textures. The dataset consists of 42000 training images, 3000 validation images and 5000 test images. The images are generated from 1084 textures collected from Brodatz dataset. Our methods is trained on only 200 images from the dataset, other methods are trained on the entire training set. Our method outperforms state of the art deep learning based methods by a healthy margin despite being significantly smaller in number of parameters and training data required for training. Similar to BSDS500 dataset we resize images to $256 \times 256$ and normalise them to have zero mean and unit variance. In post-processing we simply cluster the image into 4 regions and run 20 iterations of 2 . For quantitative comparison we have provided analyses on region accuracy metric provided with the dataset, motivated from PascalVOC class accuracy metric.

# 6 CONCLUSION

We have introduced ST-DNNs, which generalize CNNs to arbitrary shaped regions by stacking layers that solve PDEs. These are relevant to segmentation since they avoid aggregation of data across segmentation regions. They are also covariant to translations and rotations, and robust to deformations. Experiments showed that ST-DNNs achieve state-of-the-art results on texture segmentation benchmarks: they outperform state-of-the-art deep networks by 20% on edge metrics and 10% on region metrics while being **3** orders of magnitude smaller and using **2** orders of magnitude less training data. Experiments also validated covariance and robustness of ST-DNNs. To show the strength of the formulation we have tested it on four datasets, which include binary and multi-region segmentation datasets.

---

[1] https://github.com/lucasb-eyer/pydensecrf
[2] https://github.com/MMFa666/Segmentation_dataset

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

## A    APPENDIX: OUTLINE

The Appendices material is divided into the following sections.

- Analytical Proofs of Covariance and Robustness [B]
- Alternative Training Method for ST-DNN With Lower Memory Requirements [C]
- Implementation Details for Training and Joint Segmentation [D]
- Additional Experimental Analysis and Results [E]

## B    ANALYTICAL PROOFS FOR COVARIANCE AND ROBUSTNESS

The proof of Theorem 1 (covariance of ST-DNN to rotation and translation) follows from basic properties of the Laplace equation Evans (1998); we state these properties in terms of our language of covariant operators, and show the proof for the convenience of the reader. We will show Theorem 2 (robustness of ST-DNN to domain deformations) as a consequence of properties of linear PDE theory.

We repeat the definition of covariant operator:

**Definition 2** *An operator $T : \mathcal{I} \to \mathcal{I}$ is **covariant** to a class $\mathcal{W}$ of transformations if*

$$T[I \circ w] = [TI] \circ w,$$

*for every $I \in \mathcal{I}$ and $w \in \mathcal{W}$.*

We show covariance of the Laplace operator, and as a consequence, the covariance of the Poisson PDE and the ST-DNN.

**Theorem 3** *The Laplacian operator $\Delta = \sum_i \frac{\partial^2}{\partial x_i^2}$ is covariant to rotations, $x \to Rx$ where $R$ is a rotation matrix.*

**Proof 1** *Let $u \in \mathcal{I}$, and let $R$ be a rotation. Consider*

$$\frac{\partial}{\partial x_i} u(Rx) = \left[ \frac{\partial}{\partial x_i} u(Rx) \right]^T \nabla u(Rx) \tag{9}$$

$$= e_i^T R^T \nabla u(Rx) = [\nabla u(Rx)]^T R e_i. \tag{10}$$

*Also,*

$$\frac{\partial}{\partial x_i}\left[\frac{\partial u}{\partial x_j}(Rx)\right] = \left[\nabla\frac{\partial u}{\partial x_j}\right]^T(Rx)Re_i \tag{11}$$

*so,*

$$\frac{\partial}{\partial x_i}\left[\nabla u(Rx)\right] = Hu(Rx)Re_i \tag{12}$$

*so ,*

$$\frac{\partial}{\partial x_i}\left[\nabla u(Rx)\right]^T = (Re_i)^T Hu(Rx). \tag{13}$$

*Thus,*

$$\frac{\partial^2}{\partial x_i^2}u(Rx) = \frac{\partial}{\partial x_i}[\nabla u(Rx)]^T Re_i = (Re_i)^T Hu(Rx)Re_i. \tag{14}$$

*Then,*

$$\Delta(u \circ R)(x) = \sum_i \frac{\partial^2}{\partial x_i^2}u(Rx) \tag{15}$$

$$= \sum_i (Re_i)^T Hu(Rx)Re_i = tr[R^T Hu(Rx)R] \tag{16}$$

$$= tr[Hu(Rx)] \tag{17}$$

$$= \Delta u(Rx), \tag{18}$$

*where the second last equality is due to the invariance of the trace to similarity transformations.*

**Theorem 4** *The Laplacian operator is covariant to translations, $x \to x + t$, where $t$ is a vector.*

**Proof 2** *We have,*

$$\frac{\partial}{\partial x_i}[u(x+t)] = \frac{\partial u}{\partial x_i}(x+t). \tag{19}$$

*Similarly,*

$$\frac{\partial^2}{\partial x_i^2}[u(x+t)] = \frac{\partial}{\partial x_i}\left[\frac{\partial u}{\partial x_i}(x+t)\right] \tag{20}$$

$$= \frac{\partial^2 u}{\partial x_i^2}(x+t) \tag{21}$$

*Thus, $\Delta(u \circ T)(x) = (\Delta u) \circ T$ where $T(x) = x + t$.*

**Corollary 1** *The solution $u$ of the Poisson equation, i.e.,*

$$u(x) - \alpha\Delta u(x) = I(x), \tag{22}$$

*$u = T[I]$ is covariant to translations and rotations.*

**Proof 3** *This follows from the covariance of the Laplacian, and the identity map.*

We can now show covariance of the ST-DNN to translations and rotations.

**Theorem 5** *The ST-DNN equation 3 is covariant to translations and rotations.*

**Proof 4** *Since ST-DNNs are composition of solutions of Poisson Equations with fully connected layer across channels and non-linearity in multiple layers. A linear combination of channels and point-wise non-linear operation preserves the covariance of rotation and translations, hence the ST-DNNs are covariant to translations and rotations.*

We now show robustness of the ST-DNN to domain deformations with respect to the Sobolev norm; that is, we show that the output of a ST-DNN layer does not change much if deformed by a transformation with small Sobolev norm (a smooth transformation that has small displacement). The Sobolev measures the smoothness of the deformation, i.e., the $\mathbb{L}^2$ norm of the deformation and the gradient of the deformation. We have the following theorem:

**Theorem 6** *The solution of the ST-DNN is robust to deformations, i.e.,*

$$|T[I \circ w] - T[I]| \le C\|w - id\|_{H^1},\tag{23}$$

*where $T$ is the mapping from the image to the solution of the ST-DNN, $w : \Omega \to \Omega$ is a domain deformation, id is the identity map, and $H^1$ indicates the Sobolev norm.*

**Proof 5** *This is a consequence of Lemma equation 1 below, which shows that each layer of the ST-DNN is robust. Stacking such layers preserves the robustness by applying Lemma 1 successively.*

We now prove robustness of layers of ST-DNN. For convenience in the proof, we assume the input operates on the domain $\Omega = \mathbb{R}^2$, which avoids having to consider the boundary and more complicated formulas that do not affect the essence of the argument. Let

$$f[I] = r[(L \circ D \circ K) * I]\tag{24}$$

be a layer of ST-DNN. Here $r$ is the rectified linear unit, $K$ is the kernel representing, the Green's function of the Poisson equation (we assume for this proof that the domain of the image is all of $\mathbb{R}^2$), $D$ is the derivative operator representing oriented gradients (or parial derivatives of an finite order), and $L$ is a weight matrix of fully connected layer across channels.

We state the robustness of a layer of ST-DNN:

**Lemma 1** *A layer, $f[I] = r[(L \circ D \circ K * I]$, of a ST-DNN is Lipschitz continuous with respect to diffeomorphisms in the Sobolev norm, i.e.,*

$$|f[I \circ w] - f[I]| \le C\|w - id\|_{H^1},\tag{25}$$

*where $id(x) = x$ is the identity map, and $C$ is a constant (independent of $w$ and only of function of the class of images), and $\|w\|_{H^1}^2 = \int_\Omega (|w(x)| + |\nabla w(x)|^2)dx$. Note that $w - id$ is the displacement.*

**Proof 6** *Let $w$ be a smooth diffeomorphism. Then by Lipschitz continuity of the ReLu,*

$$|f[I \circ w](x) - f[I](x)| \le |M * (I \circ w)(x) - M * I(x)|\tag{26}$$

*where $M = L \circ D \circ K$. Note that by a change of variables,*

$$M * I(x) = \sum_y M(x - w(y))I(w(y)) \det \nabla w(y).\tag{27}$$

*Note that the determinent of the Jacobian appears if we weight the sum by the area measure, which approximates the integral. Therefore,*

$$M * (I \circ w)(x) - M * I(x) = \sum_y [M(x - y) - M(x - w(y)) \det \nabla w(y)]I(w(y)).\tag{28}$$

*We let $w(y) = y + v(y)$. This gives us*

$$\det \nabla w(y) = 1 + div(v(y)) + \det \nabla v(y).$$

*We may bound the second term as*

$$|div(v(y)) + \det \nabla v(y)| \le C_1 |\nabla v(y)|^2$$

*by basic inequalities. Therefore,*

$$|M(x - y) - M(x - w(y)) \det \nabla w(y)| \le$$
$$|M(x - y) - M(x - w(y))| + C_1 M(x - w(y))|\nabla v(y)|^2.\tag{29}$$

*By Lipschitz continuity of the Poisson kernel and derivatives, we have*

$$|M(x-y) - M(x - w(y))| \le C_G \|L\|_\infty |v(y)|. \tag{30}$$

*Note that the Poisson kernel has a singularity at the origin, so the statement is not precise; however, as common in PDE analysis, as we will below compute integrals of the left hand quantity, we can break the integral into two terms one that integrates the singularity in a small ball (which is finite) and the other that integrates the right hand side, that we analyze below. The former will disappear to zero as the ball goes to zero. We omit the details to avoid hiding the main argument.*

*We also have that*

$$|M(x - w(y))| \le C_2 \|L\|_\infty. \tag{31}$$

*Therefore,*

$$|f[I \circ w](x) - f[I](x)| \le C\|L\|_\infty \|I\|_\infty \int_\Omega (|v(y)|^2 + |\nabla v(y)|^2) dy \tag{32}$$

$$= C\|L\|_\infty \|I\|_\infty \|w - id\|_{H^1}^2. \tag{33}$$

For a multi layer multiple layers network with $N$ layers we will have:

$$|T[I \circ w] - T[I]| \le (\prod_{i=1}^{N} C_i \|L_i\|_\infty) C_0 \|L_0\|_\infty \|I\|_\infty \|w - \mathrm{id}\|_{H^1}^2 = C\|w - \mathrm{id}\|_{H^1}, \tag{34}$$

## C    ALTERNATIVE TRAINING METHOD FOR ST-DNN

In this section, we provide an alternative algorithm for training ST-DNN that has a smaller memory footprint than the method present in the main paper to avoid having to train on small images. Unfortunately, this is not supported in current deep learning packages, so it requires a custom implementation, and it may be more computationally costly depending on the size of the image. The basic idea is instead of back propagating (through large matrices to solve the PDE, and storing matrix multiplies) to determine derivatives of the loss with respect to the weights, we instead forward propagate the derivatives of each layer through variations of the rest of the layers (see Figure 6). The advantage of this method is that the variations of the layers with PDEs can be computed by solving the PDE itself with input being the layer weight derivative. This can be solved with an iterative technique. This does not require accumulating matrix multiplies consuming large memory as standard back propagation, as forward propagating through the PDE is analogous to a matrix-vector multiply, which as solved through an iterative technique and does not need to store the matrix explicitly.

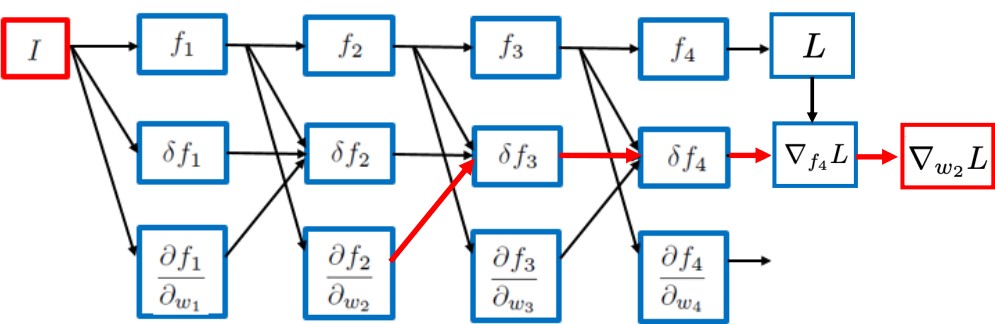

Figure 6: Example loss gradient with respect to weights calculation for the ST-DNN using the "forward propagation" method. The example red path in arrows represents the calculation of the gradient of the loss with respect to the weights of the second layer. This procedure avoids having to accumulate large matrix multiplies (to solve PDEs) as in back-propagation.

Below, we present the mathematics to show that this forward propagation technique is valid for the case of PDEs as layers, and give details of the technique.

### C.1 Variation of Shape-Tailored Descriptors

The shape-tailored descriptors are given by a Poisson equation with Neumann boundary conditions.

$$\begin{cases} u(x) - \alpha\Delta u(x) = I(x) & x \in R \\ \nabla u(x) \cdot N = 0 & x \in \partial R \end{cases}, \tag{35}$$

To take the variation of the PDEs we take the variation of each partial differential equation. To find the variation of the first equation we first calculate the value of $u(x)$ when it's input is perturbed by $\delta I$

$$u_{I+\epsilon\delta I}(x) - \alpha\Delta u_{I+\epsilon\delta I}(x) = I(x) + \epsilon\delta I(x). \tag{36}$$

The variation in the direction of $\delta I$ is defined as

$$\lim_{\epsilon \to 0} \frac{u_{I+\epsilon\delta I}(x) - u_I(x)}{\epsilon} \tag{37}$$

Next, we replace this value in the equation for the variation and get

$$u_{\delta I}(x) - \alpha\Delta u_{\delta I}(x) = \delta I(x). \tag{38}$$

Similarly for the second term, we have

$$\nabla u_{\delta I}(x) \cdot N = 0. \tag{39}$$

So the variation of the shape-tailored descriptor denoted by $u_h(x) = \delta u.\delta I(x)$ in the direction of a perturbation $\delta I(x)$ of the input image $I(x)$ is given by:

$$\begin{cases} u_h(x) - \alpha\Delta u_h(x) = \delta I(x) & x \in R \\ \nabla u_h(x) \cdot N = 0 & x \in \partial R \end{cases}, \tag{40}$$

### C.2 Derivative of Energy with respect to Weights

First, we find the variation of a layer of the shape-tailored network. A layer of shape-tailored network in composed of $r \circ L \circ T$. The expression for variation of a layer of the network is given below:

$$\begin{aligned} \delta f(I).\delta I &= (r \circ L)'(T(I))\delta T(I).\delta I \\ &= (r \circ L)'(T(I))T[\delta I] \end{aligned} \tag{41}$$

where,

$$(r \circ L)'(T(I)) = r'(L * T[I]) * L \tag{42}$$

Next, we calculate the variation through the entire network.

$$\begin{aligned} \delta F(I).\delta I =& s'\delta[f_m \circ f_{m-1} \circ ....f_1].\delta I \\ =& s'\delta f_m(\tilde{F}_{m-1}(I)) \circ \delta f_{m-1}(\tilde{F}_{m-2}(I))... \circ \delta f(I).\delta I \\ =& (s \circ r \circ L_m)'(T[\tilde{F}_{m-1}]) \circ \delta T.(r \circ L_{m-1})'(T[\tilde{F}_{m-2}]) \\ & \circ \delta T.(r \circ L_{m-2})'(T[\tilde{F}_{m-3}]) \quad ... \circ \delta T.(r \circ L_1)'(T[I]) \circ (T[\delta I]) \end{aligned} \tag{43}$$

where, $\tilde{F}_i$ is the output of the $i^{th}$ layer of the network.

Now, we derive the expression for derivative of the energy w.r.t to the weights of the network. These derivative would be used to update the weights of the network during the application of the stochastic gradient decent to the energy function.

To start with, we derive the derivative of the descriptor w.r.t. the weights of the network ($w_i$ are weights $b_i$ are biases of the fully connected layer). The derivative w.r.t. to the weights of the last layer of the network are:

$$\begin{aligned} \frac{\partial F(x)}{\partial w_m} =& s'.\frac{\partial f_m}{\partial w_m} \\ =& s'(r \circ L_m \circ T[\tilde{F}_{m-1}(x)]) \circ r'(\tilde{F}_m(x)) \circ \tilde{F}_{m-1}(x) \end{aligned} \tag{44}$$

$$
\begin{aligned}
\frac{\partial F(x)}{\partial b_m} &= s' . \frac{\partial f_m}{\partial b_m} \\
&= s'(r \circ L_m \circ T[\tilde{F}_{m-1}(x))]) \circ r'(\tilde{F}_m(x))
\end{aligned}
\tag{45}
$$

where $\tilde{F}_i$ represent the output of the network at the $i^{th}$ layer. Next, we calculate the derivative of $F$ w.r.t to $w_i$ and $b_i$, where $w_i$ and $b_i$ are the weights and biases of the fully connected layers, and $i$ represents intermediate layer of the network.

$$
\begin{aligned}
\frac{\partial F(x)}{\partial w_i} &= s' \circ \delta f_m \circ \delta f_{m-1} \circ \delta f_{m-2} ... \circ \delta f_{i+1} . \frac{\partial f_i}{\partial w_i} \\
&= (s \circ r \circ L_m)'(T[\tilde{F}_{m-1}]) \circ \delta T.(r \circ L_{m-1})'(T[\tilde{F}_{m-2}]) \\
&\quad \circ \delta T.(r \circ L_{m-2})'(T[\tilde{F}_{m-3}]) \quad ... \circ \delta T.(r \circ L_{i+1})'(T[\tilde{F}_i]) \circ r'(\tilde{F}_i(x)) \circ \tilde{F}_{i-1}(x)
\end{aligned}
\tag{46}
$$

similarly,

$$
\begin{aligned}
\frac{\partial F(x)}{\partial b_i} &= s' \circ \delta f_m \circ \delta f_{m-1} \circ \delta f_{m-2} ... \circ \delta f_{i+1} . \frac{\partial f_i}{\partial b_i} \\
&= (s \circ r \circ L_m)'(T[\tilde{F}_{m-1}]) \circ \delta T.(r \circ L_{m-1})'(T[\tilde{F}_{m-2}]) \\
&\quad \circ \delta T.(r \circ L_{m-2})'(T[\tilde{F}_{m-3}]) \quad ... \circ \delta T.(r \circ L_{i+1})'(T[\tilde{F}_i]) \circ r'(\tilde{F}_i(x))
\end{aligned}
\tag{47}
$$

Finally,

$$
\begin{aligned}
\partial_{w_i} E(I) = &\sum_i < \frac{1}{|R_i|} \int_{R_i} 2(\mathbf{F}(x) - a(x)), \frac{\partial \mathbf{F}(x)}{\partial w_i} >_{L^2} dx \\
&- \sum_i < \frac{1}{|R_i|} \int_{R_i} 2(\mathbf{F}(x) - a(x)), \int_{R_i} \frac{1}{|R_i|} \frac{\partial \mathbf{F}(y)}{\partial w_i} >_{L^2} dy dx \\
&- \sum_i \sum_{j \neq i} < 2(\mathbf{a_i} - \mathbf{a_j}), \int_{\mathbf{R_i}} \frac{\mathbf{1}}{|\mathbf{R_i}|} \frac{\partial \mathbf{F(x)}}{\partial \mathbf{w_i}} >_{\mathbf{L^2}} \mathbf{dx} \\
&+ \sum_i \sum_{j \neq i} < 2(\mathbf{a_i} - \mathbf{a_j}), \int_{\mathbf{R_i}} \frac{\mathbf{1}}{|\mathbf{R_i}|} \frac{\partial \mathbf{F(x)}}{\partial \mathbf{w_i}} >_{\mathbf{L^2}} \mathbf{dx}
\end{aligned}
\tag{48}
$$

where $< ., . >_{L^2}$ represent the inner produce of two vectors. Now, that we have the derivatives of the energy w.r.t. the weights of the network, we can train the shape-tailored networks for applications like segmentation (see Figure 6 for a visualization of the forward pass for gradients calculation).

## D  IMPLEMENTATION DETAILS

In this section we outline the important implementation details for our experiments. Figure 7 shows a schematic of the ST-DNN we use in our experiments.

### D.1  ST-DNN SETUP

We setup two versions of ST-DNN, one with MatConvNet where we use a C++ wrapper for the solution of Poisson equation of Eq 35 and a faster version in Pytorch. In the MatConvNet version the Poisson equation with Neumann boundary condition is solved with conjugate gradient algorithm, and in Pytorch version, we explicitly invert the right hand side of Equation 35 to find the solution. We noticed that training by solving the PDEs with (conjugate gradient algorithm on full size images) MatConvNet was slow compared to the faster PyTorch version, and it didn't have support available in PyTorch and TensorFlow and the results for the faster Pytorch solution (trained on downsampled images) were almost on par with the MatConvNet version (at least for the datasets we have tested on), however, learning ST-DNNs from full size images might be helpful in other applications and

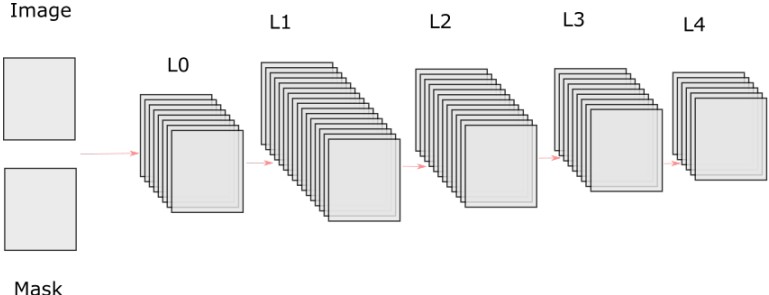

Figure 7: **Schematic of the ST-DNN:** We have a four layer ST-DNN with an additional pre-processing layer. The pre-processing layer, $L0$ in the schematic, extracts 3 color channels a gray scale channel and 4 oriented gradients at 5 scales $\alpha = \{5, 10, 15, 20, 25\}$ . The subsequent layers of the network have a smoothing layer $\alpha = \{5\}$, a fully connected layer, and a non-linearity. The number of hidden units for each layer are $100, 40, 20, 5$ respectively.

future work. The two methods are different in how they handle the solution of the PDEs. Below we provide the details of both methods.

As a first step, we provide the discretisation of the Poisson Equation 35:

$$u(i,j) - \alpha \cdot \sum_{k,l \in \mathcal{N}(i,j) \cap R} [u(k,l) - u(i,j)] = I(i,j), \quad \text{for } (i,j) \in R \qquad (49)$$

where $\mathcal{N}(i,j)$ represents the 4-pixel neighborhood of pixel $(i,j)$, which represents the $i^{\text{th}}$ row and $j^{\text{th}}$ column, and intersection means that only neighbors in the region are considered as implied by the Neumann boundary condition in the PDE, avoiding aggregation outside $R$.

For small images, we can linearise the image and write the above equations as:

$$\mathbf{A_R u} = \mathbf{I} \qquad (50)$$

where, $\mathbf{u}$ and $\mathbf{I}$ are linearised image and descriptor respectively. We can then find the solution by simply inverting the system of equations

$$\mathbf{u} = \mathbf{A_R}^{-1} \mathbf{I} \qquad (51)$$

Notice that size of $\mathbf{A_R}$ is $(m*n, m*n)$, where $m$ and $n$ are number of rows and columns in the image respectively and $\mathbf{A_R}$ depends of the mask of region $\mathbf{R}$. For small images ($32 \times 32$) the size of $\mathbf{A_R}$ is ($256 \times 256$) and we can store and invert it effectively, hence we use small images for training in our Pytorch implementation. Also, notice that if we implement the PDEs solution as in Equation 51, Pytorch (autograd) takes care of the back-propagation through the PDE layers and we do not have to explicitly implement the back-propagation (see Algorithm 1).

---

**Algorithm 1** Training of ST-DNNs

---

1: Input: Image I and ground truth mask $\mathbf{R}$
2: compute: $\mathbf{A}^{-1}$ defined in Equation equation 51. (Notice A depends on only $\mathbf{R}$)
3: initialise weights of the ST-DNN $F(x) = s \circ L_m \circ \mathbf{A}^{-1} \circ r \circ L_{m-1} \circ \mathbf{A}^{-1} \circ ... \mathbf{A}^{-1} r \circ L_0 \circ I(x)$
4: **repeat**
5:     Update Weights: using gradients from backpropagation
6: **until** validation error is minimized

---

For full size images we had to implement the equations of Section C, where we want to solve the PDEs without explicitly storing the smoothing kernel for each pixel in the image (as it would have huge memory requirement). We can solve Equation 49 using conjugate gradient algorithm as $\mathbf{A}$ in Equation 50 is symmetric positive definite (more detail in Khan et al. (2015)). Notice however that, if we use this approach we can not use autograd algorithms of standard packages like PyTorch and TensorFlow because Equation 50 is the relations from output to input (and not from input to

output), so we have to implement the gradient decent equations by calculating equations of Section C explicitly. We have implemented this using MatConvNet, however, we see that the results (at least on the datasets we have tested on) of training on small scale images are on par with results of full size image. Hence in the paper we have reported results from the Pytorch implementation. The full implementation, however, might be useful in other applications and future work.

## D.2 Inference (Joint Segmentation) Algorithm

We present the details of the algorithm for segmentation and joint ST-DNN descriptor updates (Algorithm 2). We use an approach analogous to level set methods, but instead evolve functions $\phi_i : \Omega \to \mathbb{R}$ that are smooth indicator functions of the regions $R_i$. The gradients of the energy with respect to the regions can be converted into an evolution of the indicator functions as shown in the algorithm. Note that $\kappa_i$ is the curvature of the boundary of region $R_i$, and can be written in terms of $\phi$.

---

**Algorithm 2** Texture Segmentation with ST-DNNs

1: Input: An initialization of $\phi_i$
2: **repeat**
3:     Set regions: $R_i = \{x \in \Omega : i = \mathrm{argmax}_j \phi_j(x)\}$
4:     Compute dilations, $D(R_i)$, of $R_i$
5:     Compute $\mathbf{F}_{\mathbf{R_i}}$ in $D(R_i)$, compute $\mathbf{a}_i = 1/|R_i| \cdot \int_{R_i} \mathbf{F}_{\mathbf{R_i}}(x)dx$.
6:     Compute band pixels $B_i = D(R_i) \cap D(\Omega \backslash R_i)$
7:     Compute $G_i = \|\mathbf{F}_{\mathbf{R_i}}(x)) - \mathbf{a}_i\|_2^2$ for $x \in B_i$. $\mathbf{F}$ is evaluated from the neural network.
8:     Update pixels $x \in D(R_i) \cap D(R_j)$ as follows:

$$\phi_i^{\tau + \Delta \tau}(x) = \phi_i^\tau(x) - \Delta\tau(G_i(x) - G_j(x))|\nabla\phi_i^\tau(x)| + \Delta\tau \cdot \beta\kappa_i|\nabla\phi_i^\tau(x)|. \qquad (52)$$

9:     Update all other pixels as

$$\phi_i^{\tau + \Delta \tau}(x) = \phi_i^\tau(x) + \Delta\tau \cdot \beta\kappa_i|\nabla\phi_i^\tau(x)|.$$

10:     Clip between 0 and 1: $\phi_i = \max\{0, \min\{1, \phi_i\}\}$.
11: **until** regions have converged

---

## E Additional Experiments and Results

### E.1 Ablation Studies

In Table 4, we show ablation studies of ST-DNN with varying number of layers and compare it against Siamese Khan & Sundaramoorthi (2018) (that takes hand-crafted shape-tailored descriptors as input and then uses a neural network in the channel dimension) on the Real-World Texture Segmentation Dataset. We test the performance against the number of layers in two settings. First, for both ST-DNN and Siamese, we compute the descriptors, choosing only one region that contains the whole image, i.e., $R_0 = \Omega$ to initially compute the descriptors. We do not update the descriptors as the regions of segmentation evolve to minimize the energy. Secondly, we update the descriptors as the regions evolve. This is done to show layering more PDE layers increases performance independent of the joint segmentation/descriptor approach. Results under both settings (first setting on the left and second setting on the right) are shown in Table 4. They show that increasing the number of layers in ST-DNN under both settings increases performance up to 4 layers, and then performance decreases (probably due to overfitting with too many parameters after 4 layers). Thus, the performance of ST-DNN is due to layering multiple (shape-tailored) spatial filtering layers, one of the contributions of this work. The fact that performance increases when updating the descriptors as the region evolves, which makes the descriptors shape-tailored to the segmentation, shows that the shape-tailored aspect of the descriptor is essential.

In comparison to Siamese (result reported on 4 layers processing channels of the shape-tailored pre-processing), with four layers ST-DNN out-performs it on all region metrics and performs on par on the contour metric. Note that this makes perfect sense, since stacking shape-tailored spatial

filtering layers smooths the data more than Siamese (which uses only one spatial filtering layer), making it slightly more difficult to localize the boundary, but the large effective spatial receptive size from layering spatial filtering in ST-DNN allows capturing regional properties better, leading to better performance on the region metrics.

Table 4: Left: Descriptors computed on whole image (without iteratively updating descriptors in Eqn. 8) with varying number of layers. Right: Descriptors computed on Region (with iteratively updating descriptors in Eqn. 8) with varying number of layers layers. Compared against Siamese Khan & Sundaramoorthi (2018).

| | without per-iteration descriptor update | | | | | with per-iteration descriptor update | | | |
|---|---|---|---|---|---|---|---|---|---|
| | Contour | Region metrics | | | | Contour | Region metrics | | |
| method (layers) | F-meas. | GT-cov. | Rand. Index | Var. Info. | method (layers) | F-meas. | GT-cov. | Rand. Index | Var. Info. |
| ST-DNN (1) | 0.22 | 0.79 | 0.8 | 0.74 | ST-DNN (1) | 0.23 | 0.8 | 0.8 | 0.72 |
| ST-DNN (2) | 0.35 | 0.81 | 0.82 | 0.72 | ST-DNN (2) | 0.38 | 0.84 | 0.84 | 0.68 |
| ST-DNN (3) | 0.47 | 0.86 | 0.86 | 0.59 | ST-DNN (3) | 0.51 | 0.89 | 0.9 | 0.5 |
| ST-DNN (4) | 0.52 | 0.9 | 0.9 | 0.45 | ST-DNN (4) | 0.64 | 0.94 | 0.94 | 0.35 |
| ST-DNN (5) | 0.51 | 0.89 | 0.89 | 0.56 | ST-DNN (5) | 0.63 | 0.94 | 0.94 | 0.37 |
| Siamese (4) | 0.53 | 0.89 | 0.89 | 0.47 | Siamese (4) | 0.65 | 0.92 | 0.92 | 0.43 |

### E.2 DEFORMATION ROBUSTNESS

We provide more detailed results from the deformation robustness experiments in the main paper. Figure 8 shows the plots of the performance of ST-DNN against SOTA deep learning methods, showing that ST-DNN outperforms SOTA DL methods in terms of robustness, and the margin of out-performance increases with increasing deformation amount (measured by the Sobolev norm). The robustness is measured through 3 metrics (GT-Cov, Rand Index and Variation of Information). The first two metrics higher values indicate more robustness, and the last lower values indicate more robustness. Figure 9 shows some visual results of the experiment.

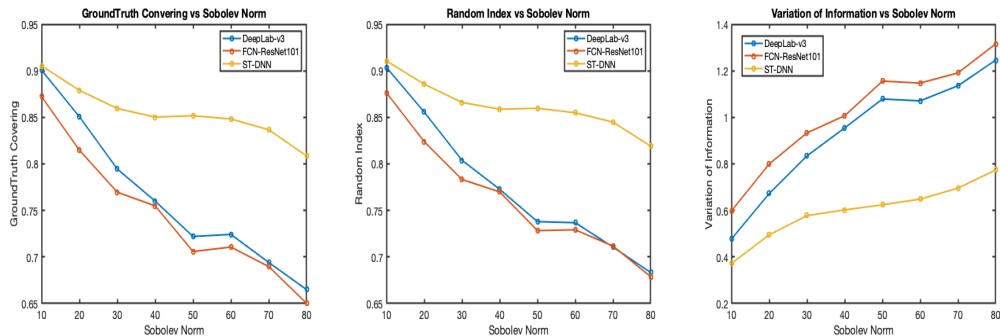

Figure 8: **Comparison of ST-DNN with SOTA deep networks for deformation robustness.** ST-DNN is more robust as measured on all metrics.

### E.3 ST-DNN PERFORMANCE AGAINST COMPETING METHODS

We show some additional qualitative samples of our results to better motivate our experiments. Figures 10 - 12, shows additional samples of results, where we compare our ST-DNN setup with state-of-the-art deep architectures on the Real-World Texture Segmentation Dataset (RWTSD). As can be seen from the samples our results are comfortably better than the SOTA deep architectures. Table 5 shows quantitative resuls on RWTSD. In our notation resnet101/deeplabv3-x-y, resnet101/deeplabv3 represents the back-bone architecture used, x represents the dataset used in training, 'd' for DUTS dataset fine-tuned on real-world texture dataset, 'm' for MSRA dataset fine-tuned on real-world texture dataset and 'all' for a combination of all datasets in Table 6 and RWTSD without any fine-tuning. With 'y' we represent the loss function used to train the network, 'ce' represents cross-entropy loss and 'ours' represents the loss function we have presented in this paper. We have tested state-of-the-art deep learning architecture with the loss introduced in this paper to show that although our loss function improves the results compared to cross-entropy, the bulk of improvement of ST-DNN over state-of-the-art networks comes from the construction of shape-tailored covariant descriptors.

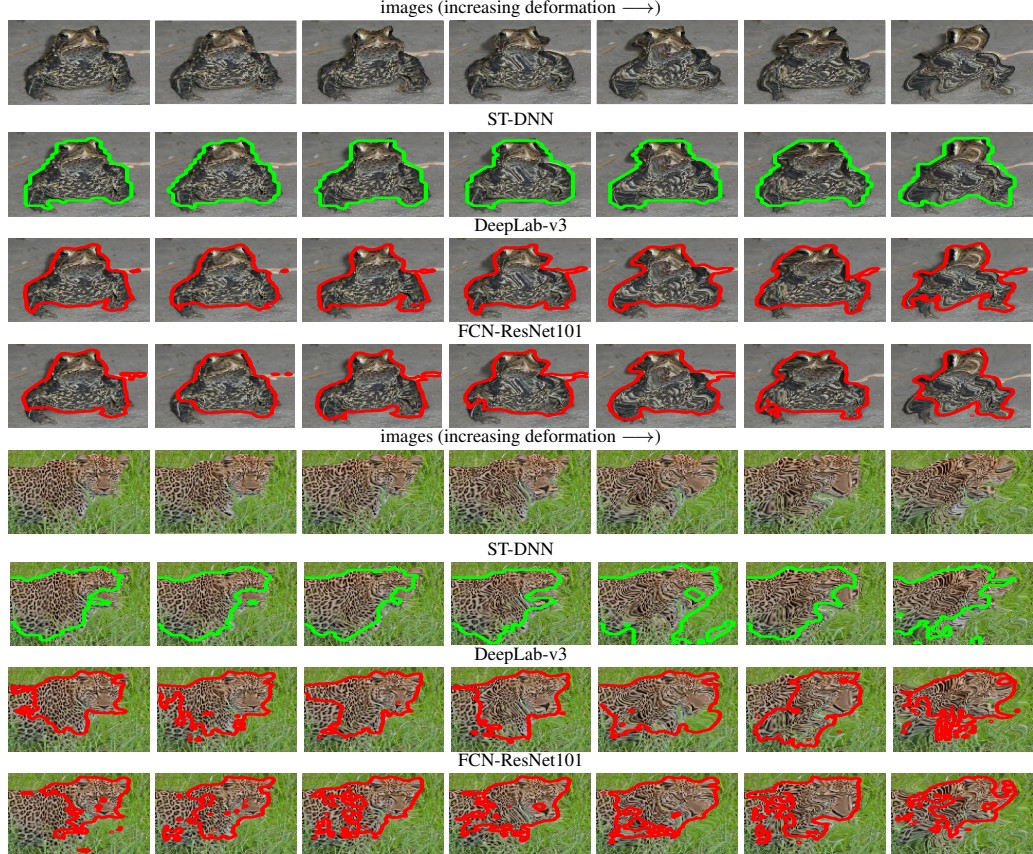

Figure 9: **Sample representative results on Real-World Texture Dataset** *with varying deformations. We compare the ST-DNNs (ours), and deep learning based methods.*

Figures 13 and 14 represent the visual samples (compared to other texture segmentation methods) on the Real-World Texture Segmentation Dataset and Synthetic Brodatz Texture Segmentation Dataset, respectively. Again, our results are better. Additional quantitative results are provided in Tables 7 (for RWTSD) and 8 (for SBTSD), comparing against more methods than in the main paper.

| | Contour | | Region metrics | | | | | |
| | F-meas. | | GT-cov. | | Rand. Index | | Var. Info. | |
| | ODS | OIS | ODS | OIS | ODS | OIS | ODS | OIS |
|---|---|---|---|---|---|---|---|---|
| ST-DNN (ours) | **0.63** | **0.63** | **0.94** | **0.94** | **0.94** | **0.94** | **0.35** | **0.35** |
| resnet100-d-ce | 0.39 | 0.39 | 0.84 | 0.84 | 0.75 | 0.75 | 0.70 | 0.70 |
| resnet-d-ours | 0.39 | 0.39 | 0.83 | 0.83 | 0.76 | 0.76 | 0.74 | 0.74 |
| resnet101-m-ce | 0.35 | 0.35 | 0.83 | 0.83 | 0.73 | 0.73 | 0.75 | 0.75 |
| resnet101-m-ours | 0.35 | 0.35 | 0.89 | 0.89 | 0.76 | 0.76 | 0.78 | 0.78 |
| resnet101-all-ce | 0.04 | 0.04 | 0.79 | 0.79 | 0.55 | 0.55 | 1.39 | 1.39 |
| resnet101-all-ours | 0.48 | 0.48 | 0.83 | 0.83 | 0.87 | 0.87 | 0.50 | 0.50 |
| resnet101-TD-ours | 0.17 | 0.17 | 0.83 | 0.83 | 0.66 | 0.66 | 0.94 | 0.94 |
| deeplabv3-all-ce | 0.42 | 0.42 | 0.86 | 0.86 | 0.75 | 0.85 | 0.66 | 0.66 |
| deeplabv3-all-ours | 0.42 | 0.42 | 0.85 | 0.85 | 0.76 | 0.76 | 0.67 | 0.67 |
| HEDXie & Tu (2015) | 0.04 | 0.04 | 0.53 | 0.53 | 0.60 | 0.60 | 1.69 | 1.69 |

Table 5: **Results on Texture Segmentation Datasets of Deep Networks**. Algorithms are evaluated on contour/ region metrics. Higher F-measure for the contour metric, ground truth covering (GT-cov), and rand index indicate better fit, and lower variation of information (Var. Info) indicates a better fit to ground truth.

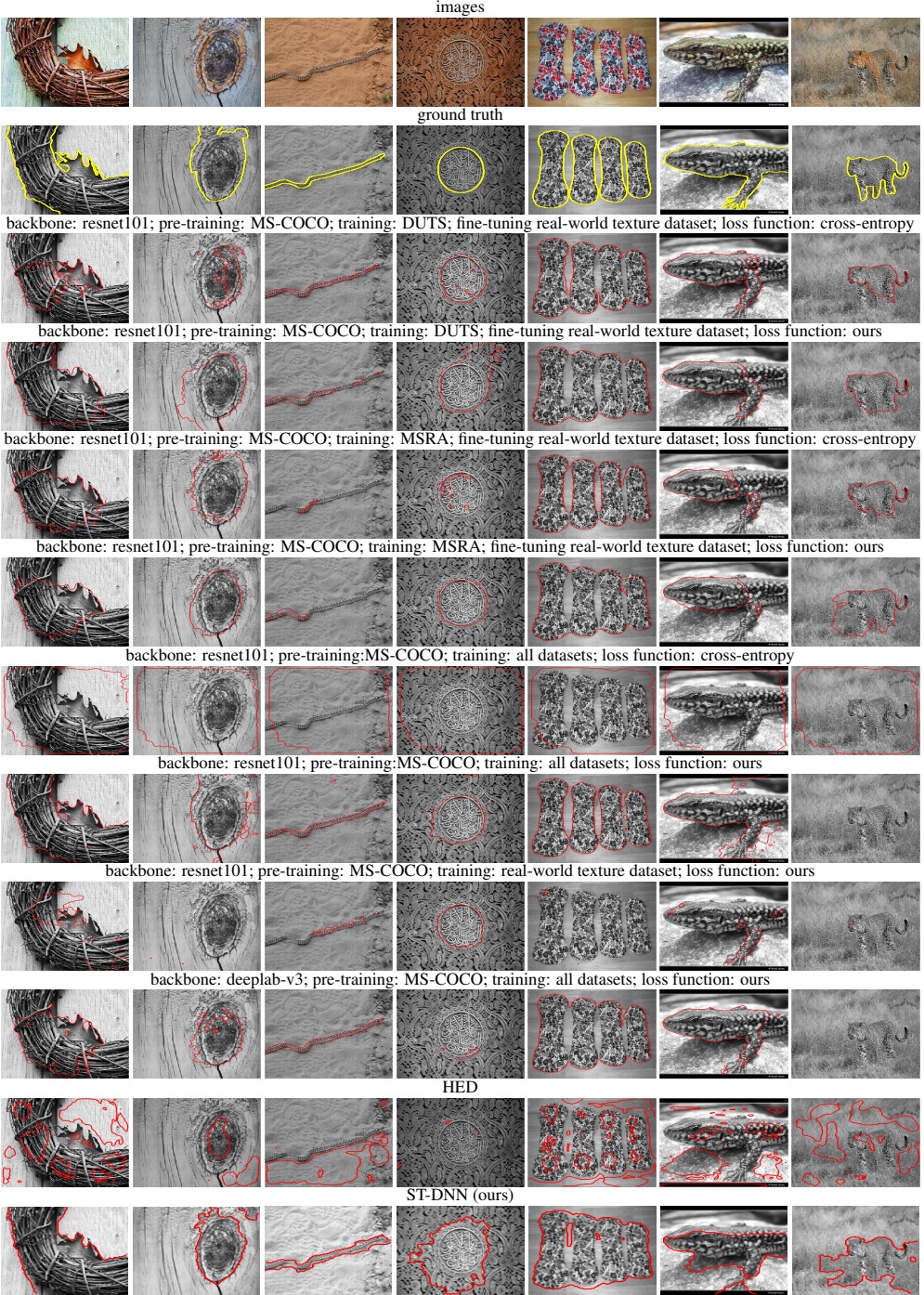

Figure 10: **Sample representative results on Real-World Texture Dataset**. *Comparison with state-of-the-art deep learning methods.*

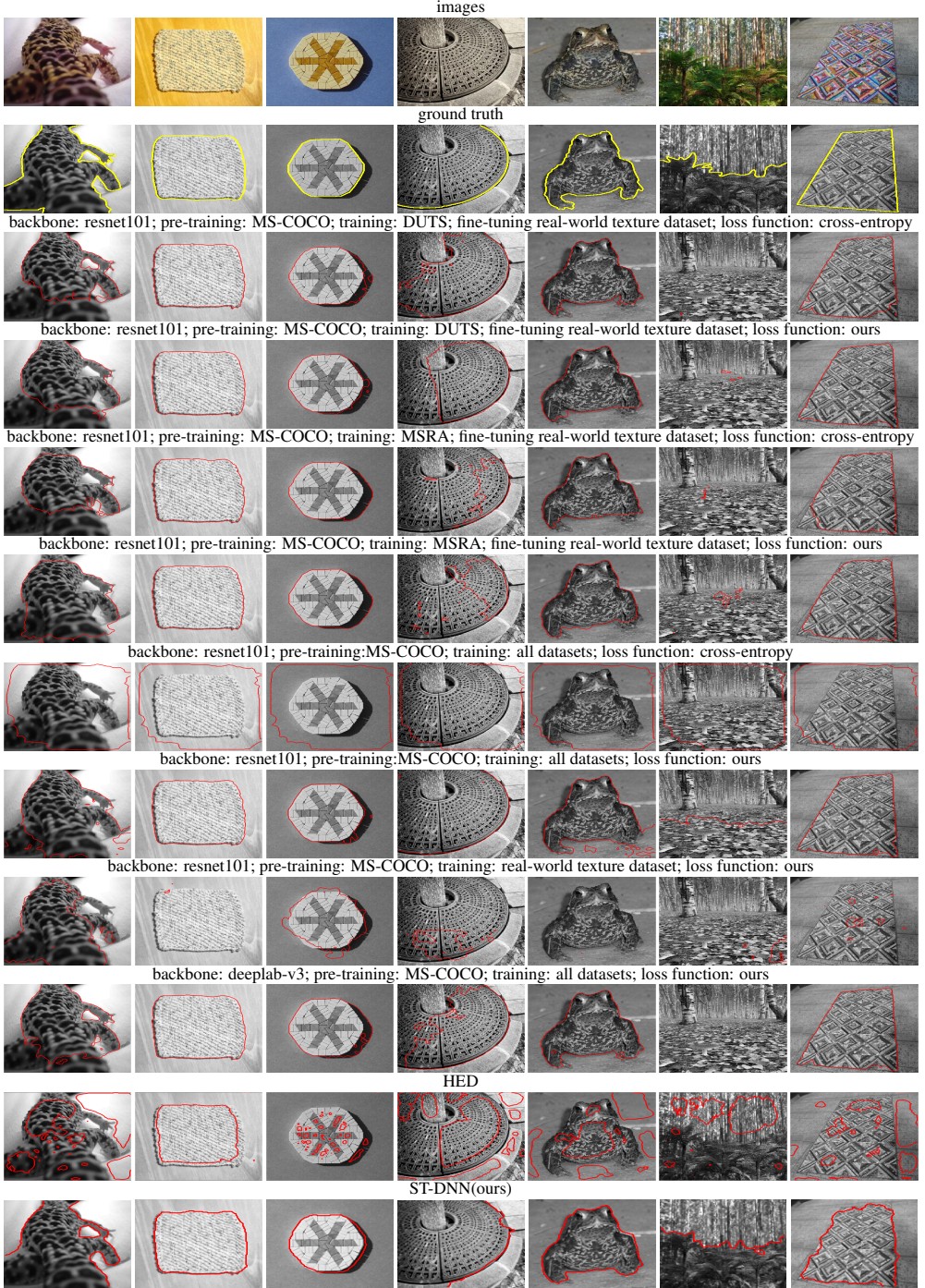

Figure 11: **Sample representative results on Real-World Texture Dataset**. *Comparison with state-of-the-art deep learning methods.*

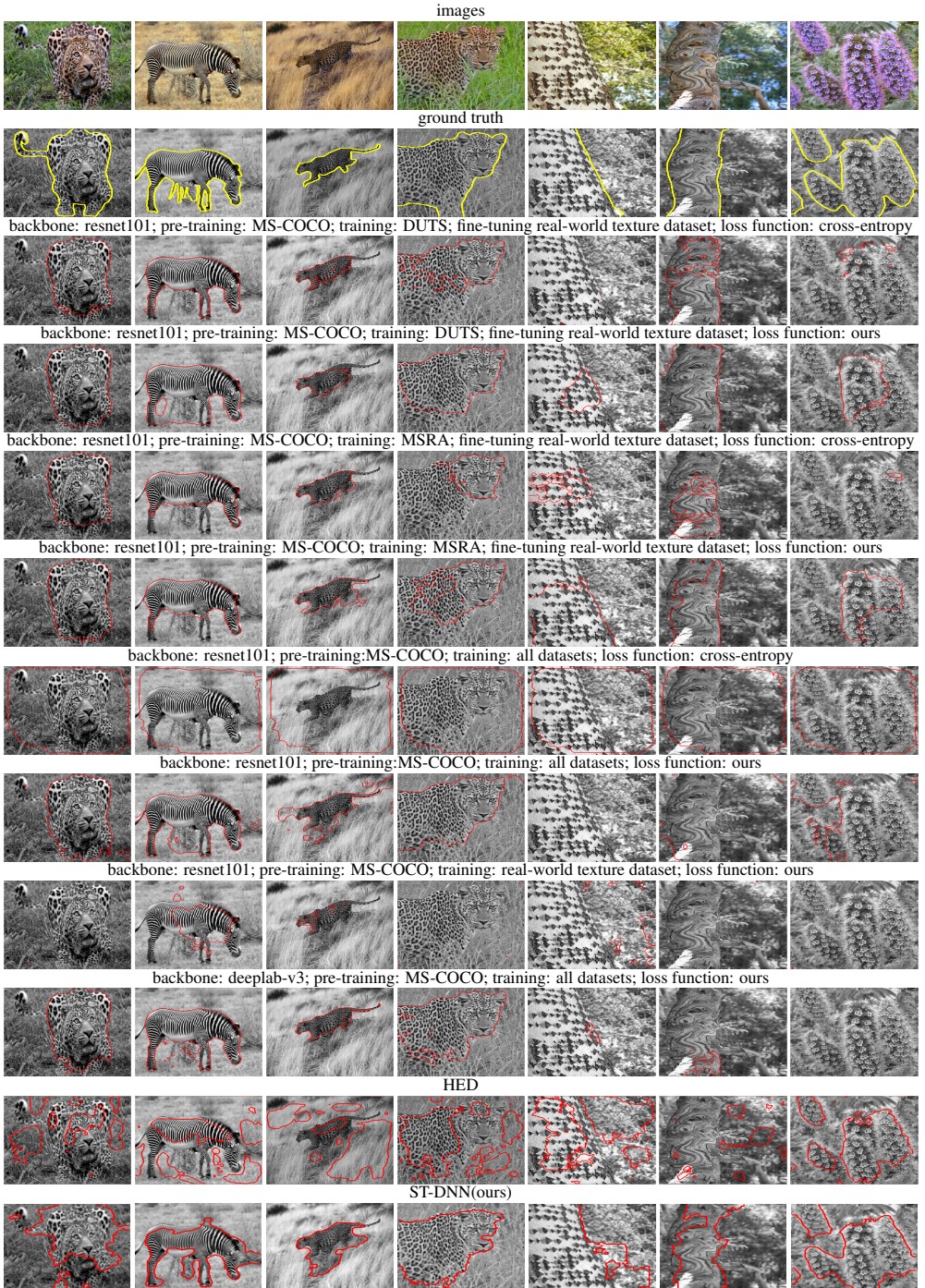

Figure 12: **Sample representative results on Real-World Texture Dataset**. *Comparison with state-of-the-art deep learning methods.*

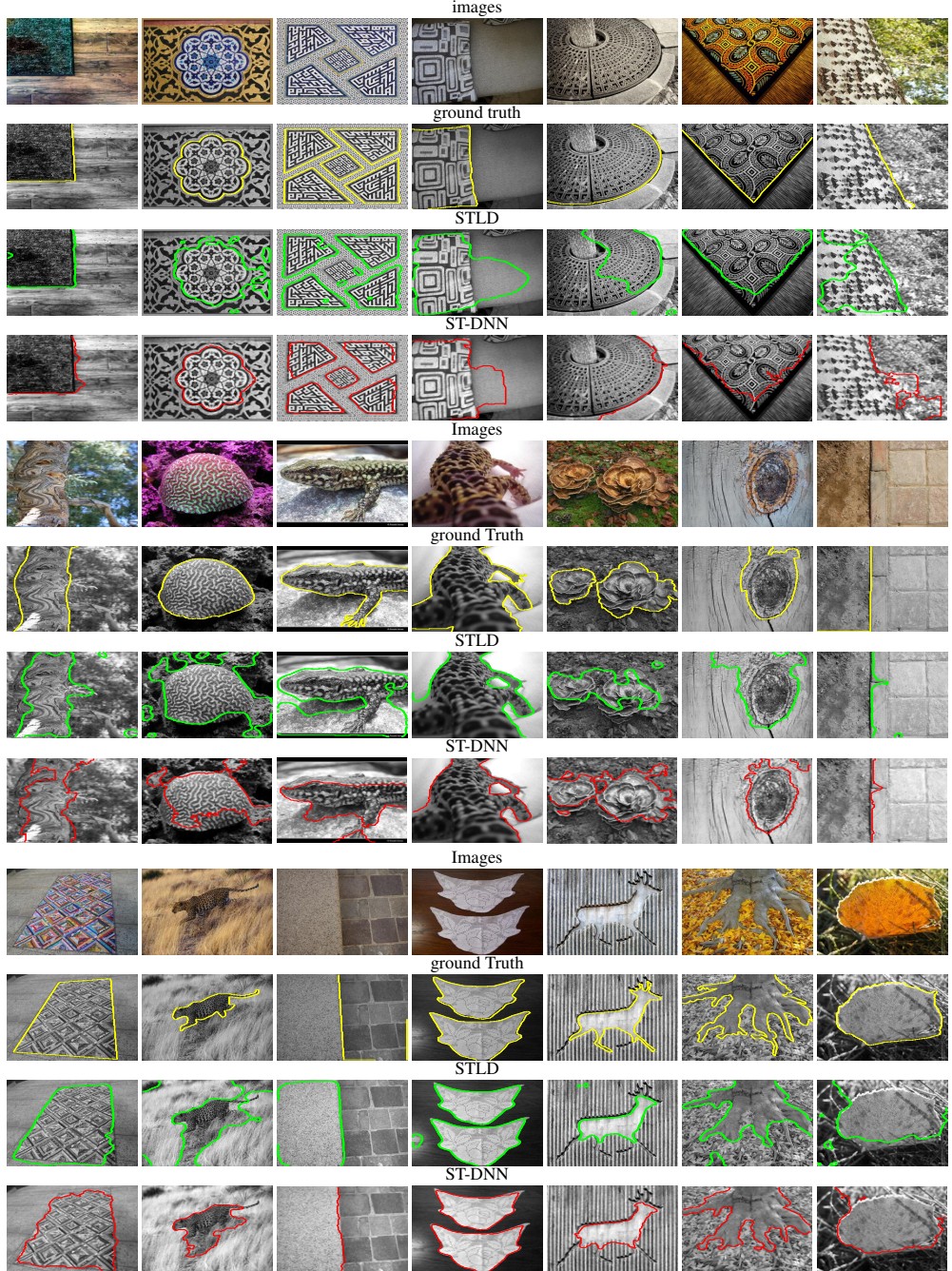

Figure 13: **Sample representative results on Real-World Texture Dataset**. *We compare the ST-DNNs (ours) and STLD.*

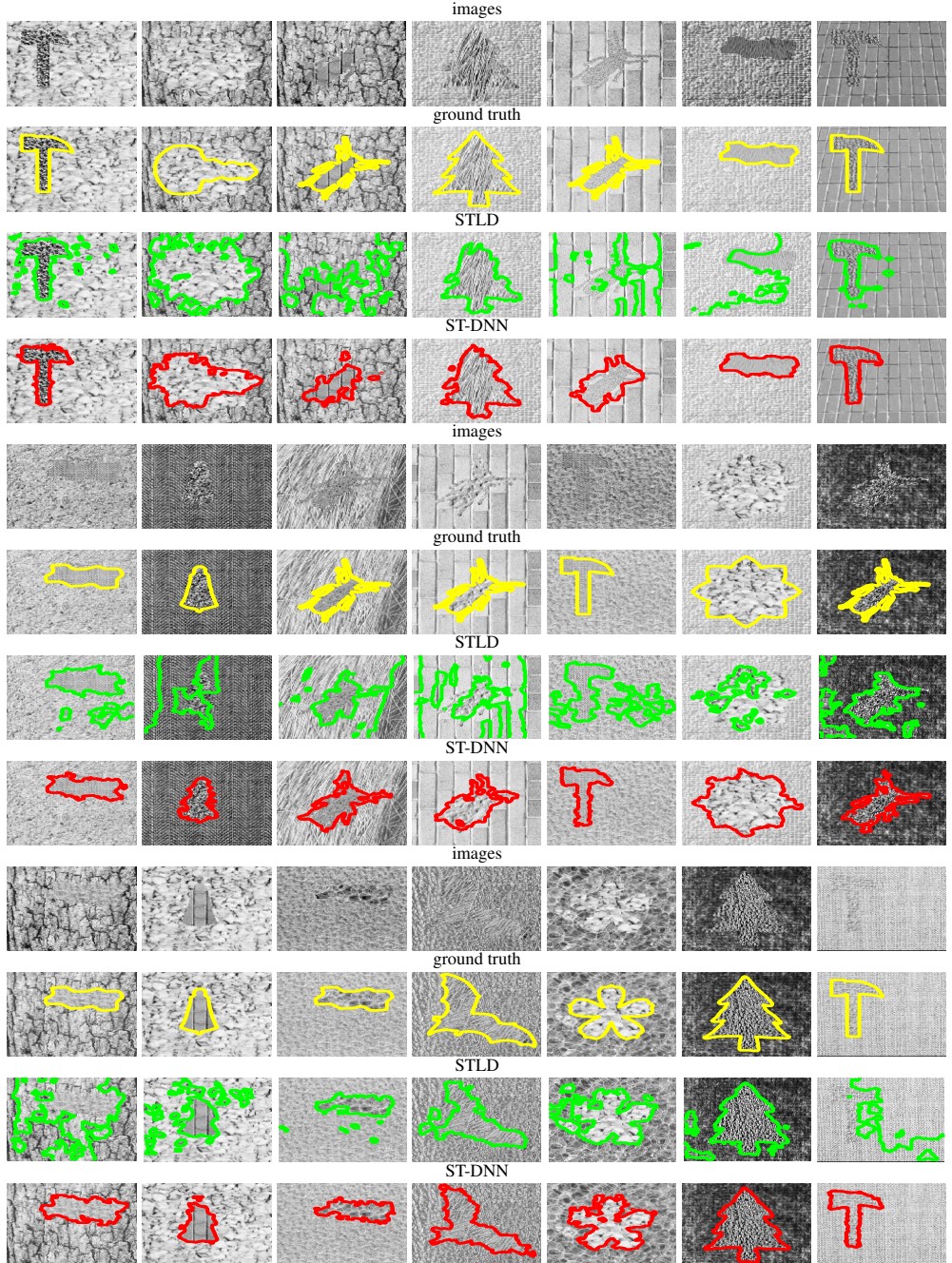

Figure 14: **Sample representative results on Synthetic Texture Dataset**. *We compare the ST-DNNs (ours) and STLD.*

| Dataset | # Images in Dataset |
|---|---|
| MSRA10k Cheng et al. (2011) | 10000 |
| DUTS-TR Zeng et al. (2018) | 10553 |
| DUTS-TE Zeng et al. (2018) | 5019 |
| ECCSD Yan et al. (2013) | 1000 |
| MSRA-B Cheng et al. (2011) | 5000 |
| PASCAL-S Li et al. (2014) | 850 |
| HKU-IS Li & Yu (2016) | 4447 |
| THUR15k Cheng et al. (2014) Yan et al. (2013) | 15000 |

Table 6: **Datasets used in training**

| | Contour | | Region metrics | | | | | |
| | F-meas. | | GT-cov. | | Rand. Index | | Var. Info. | |
| | ODS | OIS | ODS | OIS | ODS | OIS | ODS | OIS |
|---|---|---|---|---|---|---|---|---|
| ST-DNN (ours) | 0.64 | 0.64 | **0.94** | **0.94** | **0.94** | **0.94** | **0.35** | **0.35** |
| Siamese Khan & Sundaramoorthi (2018) | **0.65** | 0.65 | 0.92 | 0.92 | 0.92 | 0.92 | 0.43 | 0.43 |
| STLD | 0.58 | 0.58 | 0.86 | 0.86 | 0.88 | 0.88 | 0.63 | 0.63 |
| non-STLD | 0.20 | 0.20 | 0.83 | 0.83 | 0.84 | 0.84 | 0.79 | 0.79 |
| mcg Arbeláez et al. (2014) | 0.51 | 0.54 | 0.74 | 0.82 | 0.77 | 0.85 | 0.80 | 0.66 |
| gPb Arbelaez et al. (2011b) | 0.53 | 0.57 | 0.81 | 0.84 | 0.82 | 0.85 | 0.82 | 0.78 |
| Kok.Kokkinos (2015) | 0.64 | **0.66** | 0.56 | 0.56 | 0.56 | 0.57 | 0.92 | 0.92 |
| CB Isola et al. (2014) | 0.54 | 0.56 | 0.75 | 0.80 | 0.79 | 0.84 | 0.81 | 0.76 |
| SIFT | 0.13 | 0.13 | 0.54 | 0.54 | 0.58 | 0.58 | 1.50 | 1.50 |
| Entropy Hong et al. (2008) | 0.19 | 0.19 | 0.74 | 0.74 | 0.76 | 0.76 | 1.00 | 1.00 |
| Hist-5 Ni et al. (2009) | 0.17 | 0.17 | 0.67 | 0.67 | 0.72 | 0.72 | 1.25 | 1.25 |
| Chan-Vese Chan & Vese (2001) | 0.19 | 0.19 | 0.73 | 0.73 | 0.76 | 0.76 | 1.07 | 1.07 |
| LAC Lankton & Tannenbaum (2008) | 0.14 | 0.14 | 0.54 | 0.54 | 0.58 | 0.58 | 1.51 | 1.51 |
| Global Hist Michailovich et al. (2007) | 0.14 | 0.14 | 0.66 | 0.66 | 0.68 | 0.68 | 1.16 | 1.16 |

Table 7: **Results on Texture Segmentation Datasets**. See Table 5 caption for details on the measures.

## F ADDITIONAL COMMENTS

- **Choice of Poisson Equation**: We have chosen Poisson equation in ST-DNN because it is a linear PDE and can be efficiently solved, even for large images since the matrix $A$ in $Au = I$ is symmetric positive definite we can use conjugate gradient algorithm to solve for $u$ efficiently. However, other PDEs like Heat equation can also be used with our formulation.

- **Pre-processing** The zeroth layer (pre-processing) layer of our ST-DNN extracts color, grayscale and oriented gradient channels at multiple scales. We have used this design choice because oriented gradient are shown to be important for textural appearance of segments Khan et al. (2015); Sifre & Mallat (2013).

- **Case for Shape-Tailored descriptors:** In Khan et al. (2015) the authors show the effect of aggregation of statistics across region boundaries. They show a marked improvement in the performance of the descriptors by simply tailoring it to the region of interest.

| | Contour | | Region metrics | | | | | |
| | F-meas. | | GT-cov. | | Rand. Index | | Var. Info. | |
| | ODS | OIS | ODS | OIS | ODS | OIS | ODS | OIS |
|---|---|---|---|---|---|---|---|---|
| ST-DNN (ours) | **0.49** | **0.49** | **0.92** | **0.92** | **0.92** | **0.92** | **0.44** | **0.44** |
| Siamese Khan & Sundaramoorthi (2018) | 0.45 | 0.45 | 0.90 | 0.90 | 0.89 | 0.89 | 0.46 | 0.46 |
| STLD | 0.41 | 0.41 | 0.87 | 0.87 | 0.86 | 0.86 | 0.53 | 0.53 |
| non-STLD | 0.18 | 0.18 | 0.84 | 0.84 | 0.84 | 0.84 | 0.65 | 0.65 |
| gPb Arbelaez et al. (2011b) | 0.40 | 0.38 | 0.79 | 0.81 | 0.79 | 0.82 | 0.75 | 0.73 |
| CB Isola et al. (2014) | 0.30 | 0.29 | 0.75 | 0.77 | 0.76 | 0.79 | 1.09 | 1.08 |
| SIFT | 0.11 | 0.11 | 0.70 | 0.70 | 0.70 | 0.70 | 1.07 | 1.07 |
| Entropy Hong et al. (2008) | 0.13 | 0.13 | 0.75 | 0.75 | 0.75 | 0.75 | 0.91 | 0.91 |
| Hist-5 Ni et al. (2009) | 0.32 | 0.32 | 0.67 | 0.67 | 0.68 | 0.68 | 1.10 | 1.10 |
| Chan-Vese Chan & Vese (2001) | 0.19 | 0.19 | 0.72 | 0.72 | 0.72 | 0.72 | 0.95 | 0.95 |
| LAC Lankton & Tannenbaum (2008) | 0.14 | 0.14 | 0.72 | 0.72 | 0.70 | 0.70 | 1.14 | 1.14 |
| Global Hist Michailovich et al. (2007) | 0.28 | 0.28 | 0.75 | 0.75 | 0.75 | 0.75 | 0.79 | 0.79 |

Table 8: **Results on Synthetic Texture Segmentation Dataset**. See Table 5 caption for details on the measures.

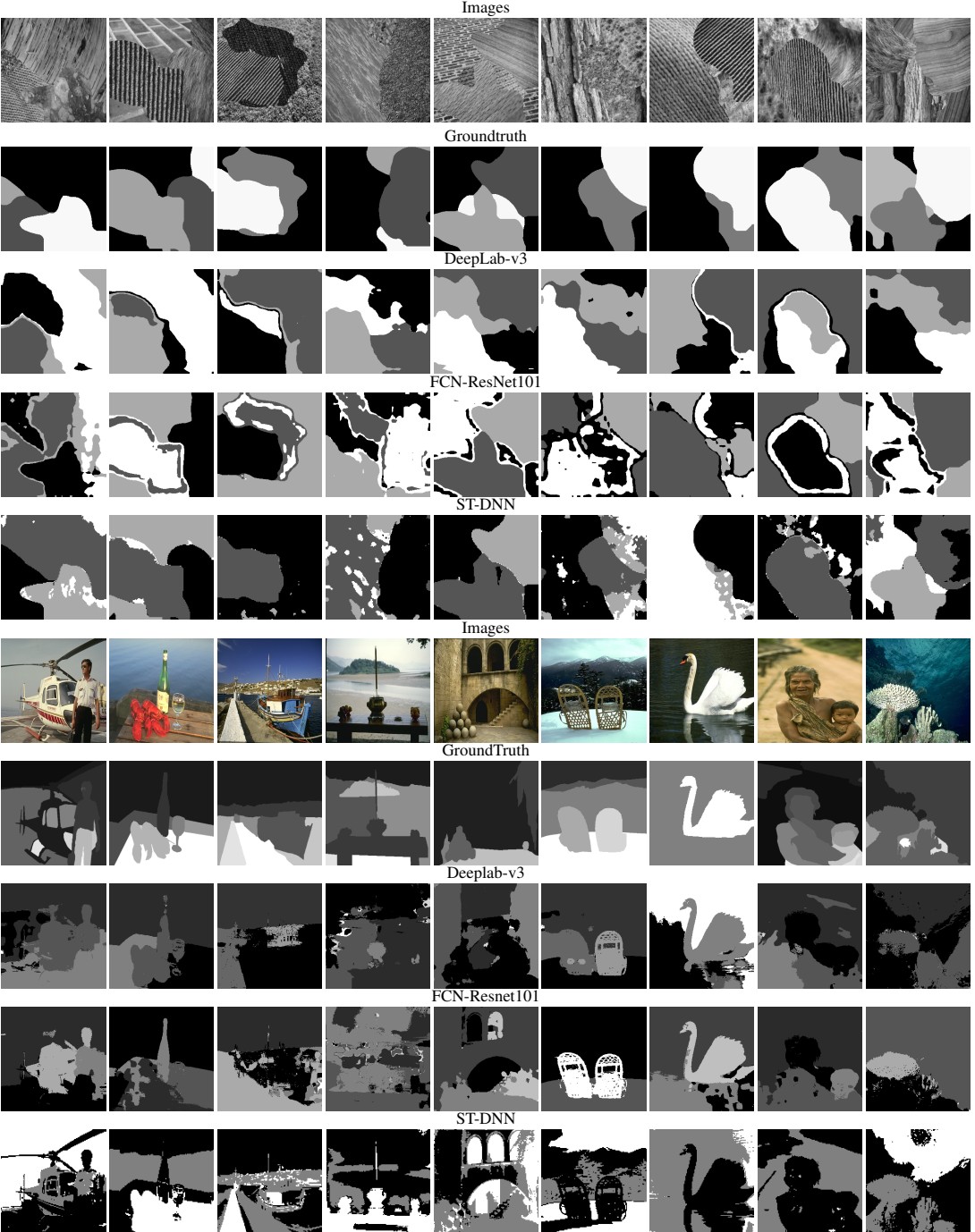

Table 9: **Results on and Synthetic multi-region Dataset and BSDS500**. Comparisons are performed against state-of-the-art deep learning based methods.

