# OpenReview forum: "Shape-Tailored Deep Neural Networks Using PDEs for Segmentation"
_ICLR.cc/2021/Conference — Reject_

### Official Review · AnonReviewer3 · 2020-10-26
**A paper with theoretical support and some good results**

**Rating:** 6
**Confidence:** 4

**Review:**

The paper proposed a new "shape-tailored" convolutional layer for improving the accuracy of semantic segmentation. The shape-tailored layer is inspired by the Poisson partial differential equation which aggregate features from neighboring pixels through the linear combinations of partial derivatives of the output of PDEs. The paper proves several properties of the proposed ST-DNN and demonstrated good results in comparisons with previous semantic segmentation algorithms.

Positive:
The paper has done good work in proving the properties of the proposed shape-tailored network and have described how to implement and train the proposed ST-DNN.

Negative:
The experimental comparisons are quite limited which it has only compared with DeepLab-v3 and FCN-ResNet101. Both methods were published on or before 2018. The more recent state-of-the-art image segmentation methods (published in 2019&2020) are not compared. The experiments on texture segmentation, especially for the experiments on increasing deformation,  are also not very impressive as these datasets are mostly synthetic.

---

> ### Author Response · Authors · 2020-11-23
> **Discussion**
>
> We would like to thank the reviewer for their useful feedback, it has helped us improve the paper a lot.
>
> To improve the experimental section of the paper, experiments on additional datasets are added to the paper. Kindly see the “Additional Experiment” comment. We agree with the reviewer that there are newer architectures for segmentation. The choice of Deeplab-v3 and FCN-Resnet is because these DNNs are based on some of the most popular architecutres and their performance is still very close to the state of the art.

---

### Official Review · AnonReviewer1 · 2020-10-27
**A clearly presented framework for shape-tailored networks, but needs better motivation**

**Rating:** 6
**Confidence:** 4

**Review:**


Summary
The paper suggests replacing convolutional layers with ST-DNNs. ST-DNNs, in contrast to conv. layers, can natively support non-rectangular neighborhoods (regions). Similarly to conv. layers, ST-DNNs can be stacked to increase expressivity. ST-DNNs themselves correspond to solutions to the Poisson PDE. The paper describes how these layers can be trained and used for prediction. Results are reported on a texture dataset.

Novelty
The authors suggest ST-DNNs that natively support non-rectangular regions. ST-DNNs are natively co-variant to translations and rotations (as opposed to CNNs)

Impact
ST-DNNs can outperform CNN-based networks in a texture segmentation setting when the data is scarce.
Clarity
The paper is exceptionally clear. The mathematical reasoning is sound and the notation helps to understand the relationship between the components. The appendix contains a plethora of additional information and is clearly structured.

Evaluation
The authors performed experiments with regard to two texture datasets, (i) Real-World Texture Segmentation, and (ii) Synthetic Texture Dataset. Both datasets are comparatively small (<300 images)

Strengths (Reasons to accept)
Mathematically speaking, the paper presents solid derivations that are easy to follow. The notations help to understand the meaning.
In general, the clarity of the plots, the diagrams and the algorithms are good
The description of the training and the prediction phase is extraordinarily clear and sensical.


Weaknesses (Reasons to reject)
The choice of the Poisson PDE seems somewhat arbitrary. The argumentation why this choice was made seems to also apply in a much more general context to simpler approaches
The authors argue (Related work) that pre-existing segmentation approaches have difficulty in handling the larger number of classes typically found in texture segmentation tasks. This claim doesn’t seem warranted and a proper justification is missing.
Section 3.2 mentions that images need inevitably to be downscaled to make the model fit into memory. This seems to be an achilles heel of the model that could have probably been addressed differently.
The evaluation of the paper is unnecessarily restricted to texture segmentation. It would have been interesting to test the system on classical segmentation problems that necessitate a more hierarchical segmentation approach (e.g. autonomous driving) as opposed to the very local nature of texture segmentation. If ST-DNNs underperform on mainstream segmentation tasks, this is valuable information to the community.
Most of the proofs are straightforward (owed to the fact that some property is inherited from the Poisson PDE) and could have either been left out or abridged
The argument that neighborhoods ideally have to be non-rectangular is not entirely convincing. A practical toy example (simpler than texture segmentation) that showcases the (supposed) shortcomings of CNNs would be useful here.
The reliance on an initial mask is a limitation of ST-DNNs compared to conventional CNNs.
Admittedly, ST-DNNs seem to cope better with small dataset sizes than CNNs in the texture setting. The readers should be convinced that this property is important, perhaps through a high-impact example where data is scarce.

Formatting
The authors did not follow the ICLR style for citations as described in Section 4 of the template, replacing named citations (e.g. Gou et al, 2017) with a numeric citation (12). This effectively gives the authors more space for content than competing submissions.

Questions and other comments to the authors
Which are the most compelling reasons to prefer Poisson PDEs over other PDEs or even simpler approaches?


=============UPDATE==============
I am generally satisfied with the answers provided by the reviewers and I have increased my score accordingly.

---

> ### Author Response · Authors · 2020-11-23
> **Discussion**
>
> We would like to that the reviewer for the meticulous feedback, it has helped us improve the paper a lot.
>
> 1. The choice of the Poisson PDE seems somewhat arbitrary.
>
> The motivation for using Poisson PDE is as follows (see also page 3 paragraph tilted shape-tailored deep network):
> it is known that the derivative of Gaussians can approximate  convolutional kernels. The generalization of Gaussian smoothing to region of arbitrary shape is the Heat Equation, which is computationally expensive to solve. There we have chosen the Poisson equation as it is the steady state of a heat equation and it can be solved efficiently (see Appendix D for more details). We have added some explanation to the "Additional Comments " section in Supplementary material. The PDE formulation also allows us to acheive the theoretical guarantees for desirable covariance and robustness properties.
>
>
> 2. pre-existing segmentation approaches have difficulty in handling the larger number of classes
>
> This comment is sepcifically about semantic segmentation methods where the output classes are restricted to a few classes (usually around $10^2$) which is not comparable to the number of texture classes in real life.
>
> 3. Section 3.2 mentions that images need inevitably to be downscaled to make the model fit into memory.
>
> Images are downsampled to use packages like PyTorch we have also trained on full size images and the downsampling does not result in any significant performance decrease (see Appendix D).
>
> 4)	The evaluation of the paper is unnecessarily restricted to texture segmentation.
>
> We would like to highlight that the main motivation of the paper is to present the first deep learning based segmentation method that is shape tailored and has desirable invariance/ robustness properties by design. DNN based methods, although they have been shown to work on semantic segmentation datasets, do not display these properties and hence their perfromance drop in many real settings where simple transformations of objects are widely present. Our work is primarily focused on the theoretical properties with proof of concept on textured datasets which displays a lot of these transformation like domain deformation, translation, rotation and scale change.
>
> Additionaly we have  extended the experimetn section as well, kindly see the “Additional Experiments” comments.
>
>
> 5) Most of the proofs are straightforward (owed to the fact that some property is inherited from the Poisson PDE) and could have either been left out or abridged.
>
> The proofs are in the Appdendix. Our intention in highlighting these properties in the main paper is to empahsize the fact that our's is the first Deep Learning based segmentation approach that exhibits these properties.
>
>
> 6) The argument that neighborhoods ideally have to be non-rectangular is not entirely convincing.
>
> Rectangular feature aggreagate statistics from across the boundaries of objects/ regions making it difficult to localize object boundaries. This has been demonstrated in previous work e.g. Khan2015. We have added a comment to "Additional Comments" section in the supplementary material.
>
> 7) The reliance on an initial mask is a limitation of ST-DNNs compared to conventional CNNs.
>
> The method can be intitialized with the clustering of our descriptors computed on the region (R) chosen to be the entire image which gives a reasonable segmentation (with e.g artificats near the bounaries similar to existing semantic segmentation methods without the post-processing). Our method further refines the segmenation iteratively to increase the accuracy of the segmentation. This iterative refinement to improve accuracy is an advantage compared to existing DNN based approaches which rely on post-processing techniques.
>
> 8) Formatting :
>
> References style corrected.
>
> 9) Questions and other comments to the authors Which are the most compelling reasons to prefer Poisson PDEs over other PDEs or even simpler approaches?
>
> See response to comment 1).

---

### Official Review · AnonReviewer2 · 2020-10-29
**The paper proposed a new DNN - ST-DNN for segmentation with PDE. Compared with traditional CNN where fixed shape regions are described, it can describe arbitrary shape regions. Experiments are conducted on texture segmentation problem, showing good results and robustness to translation, rotation and domain deformation.**

**Rating:** 5
**Confidence:** 3

**Review:**

In general, I think this paper is well-written and easy to follow, although the English can be
further improved. But the experiments part are weak for me.
1. I see that the authors conducted two experiments on texture segmentation datasets where
both are very small datasets. Why does this happen? Small datasets usually can not tell us
statistic conclusions as random things always influence the results pretty much. Or is it because
that ST-DNN can only be applied or be useful on small datasets? If yes, this needs to be stated
and clarified in the paper.
2. The experiments are conducted on two texture segmentation datasets which are both very
small with a few hundred images. In table I, the authors compare with more general baselines,
i.e., Deeplab. However, Deeplab is proposed for large dataset which will experience overfiting on
the small datasets used by the authors. So I feel that Table 1 is not a very fair comparison and
indicates less meaning to us in terms of the performance of ST-DNN compared to general
segmentation methods.
3. If we fix the consideration inside texture segmentation, where the authors kindly compare
with a few state-of-the-art in Table 2. However, only two datasets are tested while I think &gt;4 are
sufficient for a good quality publication.
Overall, I think the motivation and novelty are good. However, the validation needs to be
enhanced.

---

> ### Author Response · Authors · 2020-11-23
> **Discussion**
>
> We would like to thank the reviewer for their valueable feedback, it has helped us improve the paper a lot.
>
> 1. I see that the authors conducted two experiments on texture segmentation datasets where both are very small datasets. Why does this happen? Small datasets usually can not tell us statistic conclusions as random things always influence the results pretty much. Or is it because that ST-DNN can only be applied or be useful on small datasets? If yes, this needs to be stated and clarified in the paper.
>
> The success of our methods on small trainingset is actually one of the strenghts of our methods. DNNs even if we train them on large scale datasets and finetune on small datasets, they fail to perform on par with our method. Also, we have added additional experiments to the paper, one on large scale multi-region texture segmentation dataset with 50,000 images.
>
>
> 2. The experiments are conducted on two texture segmentation datasets which are both very small with a few hundred images. In table I, the authors compare with more general baselines, i.e., Deeplab. However, Deeplab is proposed for large dataset which will experience over-fiting on the small datasets used by the authors. So I feel that Table 1 is not a very fair comparison and indicates less meaning to us in terms of the performance of ST-DNN compared to general segmentation methods.
>
> In order to have a fair comparison and avoid overfitting the deep learning based methods are trained on standard datasets with as many as 50,000 images. After which they are fine-tuned on our training set which is further augmented (see subsection Methods in Experiment Section).
>
> To further elaborate the strenght of our method we have also added an additional experiment with large trainingset of 42,000 images.
>
> 3.If we fix the consideration inside texture segmentation, where the authors kindly compare with a few state-of-the-art in Table 2. However, only two datasets are tested while I think >4 are sufficient for a good quality publication. Overall, I think the motivation and novelty are good. However, the validation needs to be enhanced.
>
> Experiments on more datasets added to the paper, kindly refer to "Additional Experiment" Section.

---

### Official Review · AnonReviewer4 · 2020-11-03
**A novel architecture using aggregation operator over arbitrary regions but concerns on evaluation**

**Rating:** 6
**Confidence:** 4

**Review:**

This papers presents shape-tailored deep neural networks (ST-DNN) and apply to the task of texture segmentation. ST-DNN are motivated by the prior work on shape-tailored descriptors (or smoothing) that aggregate image statistics within regions of the interest, and defined as a solutions to the Poisson PDE which balances image fidelity and smoothness. This paper applies this formulation to generalize convolutions from square-shared operations to arbitrary regions, and constructs a deep neural networks by stacking this smoothness operator with 1x1 convolutions and ReLU (ST-DNN layer), repeatedly. For texture segmentation task, ST-DNN takes the input data (input images channels, together with grayscale and oriented gradients computed at 5 scales) and initial segmentation mask, and applies several layers of ST-DNN layers to produce an updated segmentation mask. The network parameters are trained using a loss that is minimized when the descriptors within a segment are homogenous, and different from descriptors in other regions.

Strengths
+ Paper presents a novel deep network architecture using shape-tailored smoothing operations to aggregate data over arbitrary regions, thus, generalizing the convolution operation.
+ Proposed formulation also enables the ST-DNN to be covariant to the in-plane rotation and translations as well as small deformations. Theoretical justifications as well as empirical validation are included (in the appendix).
+ Proposed outperform other shape-tailored variants for texture segmentation as well as earlier works on texture segmentation.
+ Proposed operation could potentially benefit other applications such as tracking or image segmentation and hence may be of relevance to broader vision community.

Concerns
- Input preprocessing for the segmentation task: The relevance of adding the grayscale, together with oriented gradients over numerous scales seems unclear. The compositional operation (simulated by stacking of multiple layers) should implicitly compute. An analysis of different feature space and how it impacts the performance seems to be missing. This would be important for the readers attempting to generalize the proposed method to other tasks.
- Evaluation datasets: Similar to prior shape-tailored methods which were applied to benchmarks datasets of tracking and segmentation such as BSDS500, it would be helpful in the proposed method can be applied and evaluated on such benchmarks to make it easier for readers to assess the benefits of the proposed method.
- Comparison with deep segmentation methods: From the texture segmentation, it seems a natural formulation would be use Siamese or triplet networks to facilitate homogeneity of within each region and discriminativeness across regions (as is done in [23], as well as other papers such as https://ieeexplore.ieee.org/document/8545348). How is the cross-entropy loss used to train the baselines? Is each texture region in the dataset a unique label? It would help better appreciate the poor segmentation performance of these baselines.
- Model selection strategy: How are the best performing hyperparameters obtained? Table 3 shows that the performance changes significantly as the number of layers increase until 4 layers and over-fits as soon as 5th layer. Are the reported quantitative results on the testing dataset the most optimistic estimate, or were the parameters determined using a model-selection/validation dataset?
- How well does the proposed method handle scales? With standard segmentation methods, the scales are handled by explicit down/up sampling layers. How does the consideration of oriented gradients at fewer scales impact the performance?

Overall the paper presents an interesting formulation which could be useful to segmentation tasks, as well as tracking (as presented in [22]) but there are several concerns which need to be addressed.

---

> ### Author Response · Authors · 2020-11-23
> **Discussion**
>
> We would like to thank the reviewer for their valuable feedback which has helped us imporve the paper.
>
> 1. Input preprocessing for the segmentation task: The relevance of adding the grayscale, together with oriented gradients over numerous scales seems unclear. The compositional operation (simulated by stacking of multiple layers) should implicitly compute. An analysis of different feature space and how it impacts the performance seems to be missing. This would be important for the readers attempting to generalize the proposed method to other tasks.
>
> The efficacy of grayscale and oriented gradients at multiple scales have been shown in Mallat2013 and Khan2018.  More details added in the paper in Supplmentary (Additional Comments) to make it clear to the reader.
>
> 2. Evaluation datasets: Similar to prior shape-tailored methods which were applied to benchmarks datasets of tracking and segmentation such as BSDS500, it would be helpful in the proposed method can be applied and evaluated on such benchmarks to make it easier for readers to assess the benefits of the proposed method.
>
> We have added additional experiments, kindly refer to the “Additional Experiments” comment
>
> 3. Comparison with deep segmentation methods: From the texture segmentation, it seems a natural formulation would be use Siamese or triplet networks to facilitate homogeneity of within each region and discriminativeness across regions (as is done in [23], as well as other papers such as https://ieeexplore.ieee.org/document/8545348). How is the cross-entropy loss used to train the baselines? Is each texture region in the dataset a unique label? It would help better appreciate the poor segmentation performance of these baselines.
>
> Notice that we have tested deep learning methods with our loss. Our loss is an efficient formulation for metric learning. Rather than optimizing the distance of the descriptor for a pair of pixels or a triplets, our energy optimizes the distance for all pixels in the regions. Furthermore, notice that our formulation is much more time and compute efficient. We have used cross entropy loss for deep learning methods just for completeness so that it is evident that cross entropy type losses don't work in appearance learning problems. Keeping loss function the same we have shown the difference in performance is only because of the architecture differences of ST-DNN to standard deep learning methods.
>
>
>
> 4. Model selection strategy: How are the best performing hyperparameters obtained? Table 3 shows that the performance changes significantly as the number of layers increase until 4 layers and over-fits as soon as 5th layer. Are the reported quantitative results on the testing dataset the most optimistic estimate, or were the parameters determined using a model-selection/validation dataset?
>
> We selected a 4 layer network based on results on a validation set where performance starts to plateau after 4 layers. Similar trends can be observed on test set [see Section ablation study]
>
> 5.How well does the proposed method handle scales? With standard segmentation methods, the scales are handled by explicit down/up sampling layers. How does the consideration of oriented gradients at fewer scales impact the performance?
>
> Since our ST-DNN aggregates statistics at multiple scales the methods are invariant to scale change. Notice the texture datasets have sufficient scale variations and we perform very well on these datasets.

---

### Official Review · AnonReviewer5 · 2020-11-09
**Interesting method and good results on specific problem of texture segmentation**

**Rating:** 6
**Confidence:** 3

**Review:**

The authors propose shape-tailored deep neural networks which performs filtering according to Poisson PDE using convolutions and output robust descriptors for the task of texture segmentation.

The strengths:
1. The relationship to Poisson PDE and its formulation as a convolution.
2. Property of covariance illustrated through proof and experimentation
3. Exact formulation and numerical approximations to the approach for faster training
4. Extensive experiments on texture segmentation.

Weakness:
1. Focus on only texture segmentation. It will be valuable to show experments on background-foreground segmentation which is a similar task.
2. Inadequate validation of the decriptors derived from ST-DNN. If the descriptors indeed are more valuable, this should be demonstrated using simple tasks like using the 'average' or 'aggregated' descriptor for linear classification tasks. This would be valuable to the representation learning community, but was not addressed in the experiemnts.
3. ST-DNN is favourable with respect to data efficiency and parameter efficiency, but the sophisticated nature of the Poisson operator adds to the complexity. The cons in terms of flops for training and inference should be discussed. Also, the authors should clarify the limitations of the approach more clearly.

Post rebuttal comment: Having read all the reviews and in light of the additional experiments, I am slightly raising my rating. The reason I am still not fully convinced is no experiments indicating the quality of descriptors learned by the method.

---

> ### Author Response · Authors · 2020-11-23
> **Discussion**
>
> We would like to thank the reviewer for their valubale feedback. It has helped us improve the paper alot.
>
> 1. Focus on only texture segmentation. It will be valuable to show experiments on background-foreground segmentation which is a similar task.
>
> Additional Experiments added. Kindly see the “Additional Experiments” Comment
>
> 2. Inadequate validation of the descriptors derived from ST-DNN. If the descriptors indeed are more valuable, this should be demonstrated using simple tasks like using the 'average' or 'aggregated' descriptor for linear classification tasks. This would be valuable to the representation learning community, but was not addressed in the experiments.
>
> Additional Experiments added. Kindly see the “Additional Experiments” Comment above.
>
>
> 3. ST-DNN is favourable with respect to data efficiency and parameter efficiency, but the sophisticated nature of the Poisson operator adds to the complexity. The cons in terms of flops for training and inference should be discussed. Also, the authors should clarify the limitations of the approach more clearly.
>
> Poisson equation indeed adds an overhead in the computation but it is much smaller than the compute requirement of large networks. (Please see Appendix D for details)

---

### Author Response · Authors · 2020-11-23
**Additional Experiments**

We would like to thank the reviewers for their meticulous feedback. It is encouraging to see that the reviewers have taken an in-depth look at the paper and have provided valuable feedback.


The primary concern of the reviewers was about the experiment section. We had tested on two segmentation datasets proposed in Khan 2015. To address this concern we have tested on two additional datasets BSDS500 and Multi-region synthetic texture dataset [https://github.com/MMFa666/Segmentation_dataset ]. BSDS500 is one of the most widely used datasets in edge-based approach to segmentation and Multi-region synthetic texture dataset is a large scale multi-region texture segmentation dataset with 50,000 images. Our method consistently outperforms state-of-the-art methods on all datasets.

We would also like to highlight that the main motivation of the paper is to present the first deep learning based segmentation method that is shape tailored and has desirable invariance/ robustness properties by design. DNN based methods, although they have been shown to work on semantic segmentation datasets, do not display these properties and hence their perfromance drop in many real settings where simple transformations of objects are widely present. Our work is primarily focused on the theoretical properties with proof of concept on textured datasets which displays a lot of these transformation like domain deformation, translation, rotation and scale change.

---

### Decision · Program_Chairs · 2021-01-07
**Final Decision**

**Decision:**

Reject

**Comment:**

This paper proposes a new kind of CNN that convolves on deformable regions and cooperates with the Poisson equation to determine the deformable regions. Experiments on texture segmentation look promising.

Pros:
1. The paper is well written and easy to follow.
2. The idea is interesting and the reviewers liked it.
3. The experiments on texture segmentation are promising.

Cons:
1. Actually, convolution on non-rectangular region is not new, in contrast to the authors' claim and reviewers' belief, although the authors may argue that the mechanisms of determining the region for convolution are different and the CNNs are used for different tasks. See, e.g.,

Jifeng Dai, Haozhi Qi, Yuwen Xiong, Yi Li, Guodong Zhang, Han Hu, Yichen Wei: Deformable Convolutional Networks. ICCV 2017: 764-773

and other papers by the same first author. So the AC would discount the novelty of the paper.

2. As most of the reviewers commended, the experiments on texture segmentation were insufficient. Although extra experiments were added (thank the authors' effort on doing this), the reviewers actually still deemed that they were not convincing enough, e.g., should compare with more state-of-the-art methods.  Reviewers #2&#4 both confirmed this issue in confidential comments.

Although Reviewer #1 increased his/her score, the final average score is still below the threshold. So the AC decided to reject the paper.